# MER-DG: Modality-Entropy Regularization for Multimodal Domain Generalization

Yavuz Yarici [1]   Ghassan AlRegib [1]

## Abstract

Deploying multimodal models in real-world scenarios requires generalization to new environments where recording conditions differ from training, a challenge known as multimodal domain generalization (MMDG). Standard architectures employ separate encoders for each modality and a fusion module, training the system end-to-end by optimizing on the fused features. In this paper, we identify that such joint optimization causes encoders to exploit cross-modal co-occurrences, statistical relationships between modalities that arise from source-specific recording conditions, rather than learning domain-invariant features. We term this failure mode Fusion Overfitting. To address this, we propose Modality-Entropy Regularization for Domain Generalization (MER-DG), which maximizes the entropy of each encoder's feature distribution to preserve feature diversity. MER-DG is architecture-agnostic and integrates into existing multimodal frameworks as an additive loss term. Extensive experiments on EPIC-Kitchens and HAC benchmarks demonstrate average improvements of $\sim 5\%$ over standard fusion and $\sim 2\%$ over state-of-the-art methods.

## 1. Introduction

The increasing availability of multimodal data, such as audio-visual streams (Chen et al., 2020) and multi-sensor setups (Kaviani et al., 2025), has led to growing interest in multimodal learning. The core motivation is that different modalities provide complementary information, such as the visual motion of slicing vegetables paired with the rhythmic acoustic signature of a knife hitting a cutting board. This complementarity is particularly valuable under partial degradation: when one modality is corrupted, the other may preserve sufficient information for accurate prediction (Baltrušaitis et al., 2018).

A central challenge in deploying such systems is ensuring robust performance under distribution shift, a problem known as Multimodal Domain Generalization (MMDG). Models trained on data from specific source domains must generalize to target domains where recording conditions, sensor characteristics, or subjects may vary (Baek et al., 2024; Yarici et al., 2025). This capability is essential in applications such as medical imaging (Li et al., 2020), autonomous systems (Kokilepersaud et al., 2023), and video understanding (Kaviani et al., 2025).

Standard approaches to MMDG employ multimodal architectures with separate encoders for each modality and a late fusion module that aggregates their outputs, as shown in Figure 1. This system is typically trained end-to-end by minimizing a loss function such as cross-entropy on the fused features. While this joint optimization strategy achieves strong performance when training and testing data share the same distribution, we identify a systematic problem that arises from the interaction between the structure of multimodal data and the joint optimization objective.

The problem stems from the unique nature of fused features in multimodal learning, which introduces a generalization challenge not present in unimodal learning. In multimodal data, recording conditions such as sensor placement, acoustic properties, and lighting characteristics determine how signals from different modalities appear together. We refer to these patterns as cross-modal co-occurrences. These are statistical relationships between modalities that arise from source-specific recording conditions rather than from the underlying semantic content of the task. The source dependence of these cross-modal co-occurrences poses a fundamental problem for joint optimization. Since the joint optimization objective operates on fused features, the model learns to exploit cross-modal co-occurrences for prediction. Under this objective, each encoder is optimized based on its contribution to the fused representation, driving it to

[1]OLIVES at the Center for Signal and Information Processing CSIP, School of Electrical and Computer Engineering, Georgia Institute of Technology, Atlanta, GA, USA. Correspondence to: Yavuz Yarici <yavuzyarici@gatech.edu>, Ghassan AlRegib <alregib@gatech.edu>.

*Proceedings of the $43^{rd}$ International Conference on Machine Learning*, Seoul, South Korea. PMLR 306, 2026. Copyright 2026 by the author(s).

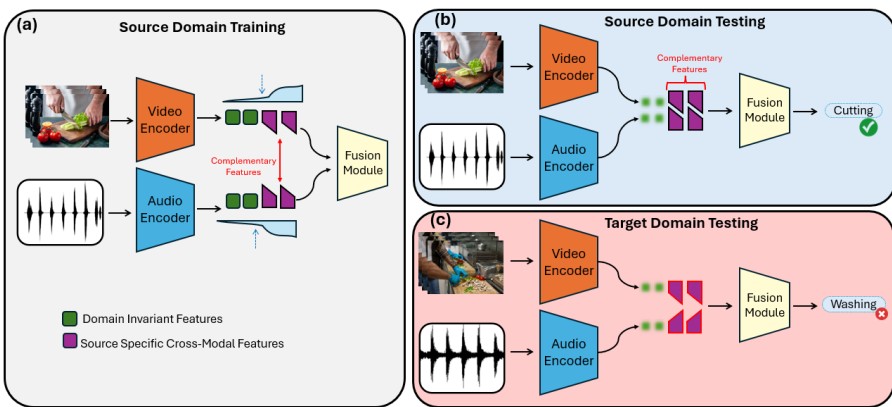

*Figure 1.* Fusion Overfitting in multimodal domain generalization. **(a)** During source domain training, the fusion objective causes encoders to weight their representations toward source-specific cross-modal features (purple) that complement each other, while domain-invariant features (green) remain underutilized. **(b)** In source domain testing, the learned cross-modal features align correctly, enabling accurate prediction. **(c)** In target domain testing, different recording conditions disrupt the learned co-occurrences. The cross-modal features no longer align, causing incorrect predictions.

produce features that are useful only when combined with the other encoder's output. As illustrated in Figure 1(a), this causes encoders to weight their representations toward source-specific cross-modal features (purple) that complement each other, while domain-invariant features (green) that could generalize independently remain underutilized.

We term this phenomenon Fusion Overfitting. Consider a model trained to recognize cutting from video and audio recorded in quiet home kitchens. Domain-invariant features exist in each modality independently: the visual pattern of a knife contacting food, or the rhythmic audio signature of chopping. However, in quiet environments, distinctive cutting sounds reliably co-occur with the visual activity. Audio amplitude spikes precisely when the knife motion is visible, and relative silence accompanies visual pauses. Rather than learning domain-invariant features, the model learns cross-modal features that exploit these source-specific co-occurrences. During source domain testing (Figure 1(b)), the learned cross-modal features complement each other, enabling correct prediction. However, when deployed to a target domain such as a noisy commercial kitchen (Figure 1(c)), background noise disrupts the learned audio-visual co-occurrences. The cross-modal features no longer align, causing incorrect predictions.

Prior MMDG methods target different aspects of multimodal learning. RNA-Net (Planamente et al., 2022) addresses the feature norm discrepancy between modalities through relative norm alignment. SimMMDG (Dong et al., 2023) decomposes representations into shared and modality-specific components. CMRF (Fan et al., 2024) flattens the multimodal loss landscape through cross-modal training. While these methods improve generalization, none directly addresses Fusion Overfitting.

To address Fusion Overfitting, we propose Modality-Entropy Regularization for Domain Generalization (MER-

DG). Our key insight is that Fusion Overfitting causes encoders to lose feature diversity: feature dimensions not needed for exploiting cross-modal patterns become inactive, and domain-invariant features are discarded in the process. By maximizing the entropy of each encoder's feature distribution, we ensure that all feature dimensions remain active, preserving domain-invariant features that would otherwise be lost. When cross-modal co-occurrences shift at test time, these preserved features enable robust prediction. Our regularization objective is architecture-agnostic and integrates into any multimodal framework with separate encoders and a fusion module.

Our contributions are:

- We identify and analyze **Fusion Overfitting**, a phenomenon in MMDG where joint optimization causes encoders to overfit to source-specific cross-modal co-occurrences rather than learning domain-invariant features.

- We propose Modality-Entropy Regularization for Domain Generalization (MER-DG), a simple and architecture-agnostic regularizer that maximizes the entropy of each encoder's feature distribution. By ensuring all feature dimensions remain active, MER-DG preserves domain-invariant features that standard fusion training would discard.

- We demonstrate consistent improvements across four multimodal baselines on EPIC-Kitchens and HAC benchmarks under both multi-source and single-source settings, with average improvements of $\sim 5\%$ on standard fusion architectures and $\sim 2\%$ on state-of-the-art MMDG methods.

- We verify that MER-DG restores standalone encoder performance to near-independent training levels, confirming that entropy regularization counteracts Fusion Overfitting.

## 2. Related Work

### 2.1. Multimodal Domain Generalization

While domain generalization has been extensively explored in unimodal settings, work on multimodal domain generalization (MMDG) is relatively limited. Early efforts such as RNA-Net (Planamente et al., 2022) identify that naive multimodal training can lead to discrepancies in characteristic norms between modalities. To address this, RNA-Net introduces a relative norm alignment loss that balances the mean feature norms of audio and visual streams. More recent methods explicitly exploit the structure of multimodal representations. SimMMDG (Dong et al., 2023) decomposes each modality into shared and specific components and combines lightweight independence and alignment regularizers with supervised contrastive learning on the shared space. CMRF (Fan et al., 2024) brings flat-minima ideas from unimodal DG to the multimodal setting, integrating sharpness-aware minimization with cross-modal training to mitigate modality competition. Similarly, MBCD (Wang et al., 2025) addresses optimization imbalance through adaptive modality dropout and collaborative distillation with an EMA-based teacher to promote convergence to flatter minima. Huang et al. (2025) argue that unimodal DG techniques transfer more effectively once modalities are embedded into a unified representation space; they disentangle domain-general and domain-specific factors within this unified embedding. These methods address optimization dynamics, representation structure, and loss landscape geometry. However, none directly addresses Fusion Overfitting, where optimizing only the fused output causes each encoder to learn features that exploit source-specific cross-modal co-occurrences rather than domain-invariant features that generalize independently.

### 2.2. Entropy Regularization

Entropy-based regularization has emerged as an effective approach for improving representation quality. In self-supervised learning (SSL), the core challenge is preventing collapse to trivial solutions in the absence of labels. Methods such as Barlow Twins (Zbontar et al., 2021), VICReg (Bardes et al., 2021), and AdaDim (Kokilepersaud et al., 2025) address this by combining an *invariance* objective, which pulls together embeddings of augmented views, with decorrelation terms that prevent informational collapse. W-MSE (Ermolov et al., 2021) applies whitening transformations directly to scatter representations uniformly. Maximum Entropy Coding (MEC) (Liu et al., 2022) formalizes this intuition through an information-theoretic lens, showing that maximizing the entropy of representations reduces bias and improves transfer to downstream tasks. These methods demonstrate that entropy maximization preserves representational richness and prevents collapse in unimodal representation learning. However, they address collapse within a single encoder arising from unimodal optimization dynamics. In contrast, Fusion Overfitting emerges from the interaction between encoders under joint optimization. By applying entropy regularization to each encoder independently, we ensure that individual encoders maintain diverse representations, preventing the complementary specialization that leads to representation collapse under fusion training.

Entropy-based objectives have also been leveraged in domain generalization with different targets. Zhao et al. (2020) use entropy regularization to learn conditional-invariant features. ADVENT (Vu et al., 2019) applies entropy minimization to classifier outputs in domain adaptation to encourage confident predictions on target domains. The Invariant Information Bottleneck (Li et al., 2021) combines invariant risk minimization with information bottleneck to filter spurious correlations. These methods target prediction entropy or domain-invariant representations. In contrast, MER-DG maximizes feature entropy within each encoder to preserve representational diversity, ensuring domain-invariant features are not discarded during fusion training.

## 3. Analysis

In this section, we first formalize the multimodal domain generalization setting and the standard fusion training objective. We then demonstrate empirically that this objective causes encoders to overfit to source-specific cross-modal features while losing domain-invariant features, resulting in degraded generalization on the target domain. Our empirical analysis examines three complementary categories of evidence: representational evidence through cross-domain alignment and direct domain classification, geometric evidence through spectral structure, and behavioral evidence through standalone encoder performance.

### 3.1. Preliminaries

We consider the Multimodal Domain Generalization (MMDG) setting with $S$ labeled source domains and an unseen target domain. Let the $s$-th source domain be denoted as

$$\mathcal{D}_{\text{src}}^{(s)} = \{(x_j^{(s)}, y_j^{(s)})\}_{j=1}^{n_s}, \quad s = 1, \ldots, S,$$

where $x_j^{(s)}$ is a multimodal input and $y_j^{(s)} \in \mathcal{Y}$ is the corresponding label. Each input sample consists of $M$ modalities $x_j^{(s)} = (x_j^{(s,1)}, \ldots, x_j^{(s,M)})$, with $x_j^{(s,m)} \in \mathcal{X}^{(m)}$ denoting the observation from modality $m$ (e.g., Video, Audio).

At deployment time, the model is evaluated on an unseen target domain with distribution $P_{XY}^{\text{tgt}}$, which is inaccessible during training. The goal of MMDG is to learn a predictive

function $f : \mathcal{X} \to \mathcal{Y}$ from the source domains $\{\mathcal{D}_{\mathrm{src}}^{(s)}\}_{s=1}^{S}$ that minimizes the expected risk on the target domain:

$$f^{\star} = \arg\min_{f} \; \mathbb{E}_{(x,y)\sim P_{XY}^{\mathrm{tgt}}}\left[\ell(f(x), y)\right], \qquad (1)$$

where $\ell(\cdot, \cdot)$ is a task loss.

We decompose the model $f$ into modality-specific encoders and a fusion module. Let $f_m : \mathcal{X}^{(m)} \to \mathcal{Z}^{(m)}$ be the encoder for modality $m$, producing a feature embedding $z_m = f_m(x^{(m)})$. These embeddings are aggregated by a fusion module $g : (\mathcal{Z}^{(1)}, \ldots, \mathcal{Z}^{(M)}) \to \mathcal{Z}_{\mathrm{joint}}$ to produce a joint representation for final classification.

Standard multimodal training optimizes the empirical risk on the joint output across all source domains:

$$\min_{\theta} \sum_{s=1}^{S} \mathbb{E}_{(x,y)\sim\mathcal{D}_{\mathrm{src}}^{(s)}}[\mathcal{L}(g(z_1, \ldots, z_M), y)], \qquad (2)$$

where $\theta$ denotes the parameters of both the encoders $\{f_m\}$ and the fusion module $g$. This objective optimizes only the joint prediction and places no constraint on individual encoder representations. As a result, encoders may overfit to source-specific cross-modal co-occurrences, producing features useful only in combination while domain-invariant features that could generalize independently are lost. We term this failure mode Fusion Overfitting and verify it empirically in the following section.

### 3.2. Empirical Analysis

We now verify empirically that the fusion training objective causes encoders to overfit to source-specific cross-modal features. We conduct our analysis on EPIC-Kitchens (Damen et al., 2018), a multimodal action recognition benchmark with video and audio modalities recorded across multiple kitchen environments. Each environment constitutes a domain, allowing us to measure cross-domain generalization. Full dataset and implementation details are provided in Section 5.1. We examine three complementary categories of evidence: representational evidence through cross-domain alignment and direct domain classification, geometric evidence through spectral structure, and behavioral evidence through standalone encoder performance.

**Source-Target Domain Representation Alignment.** We quantify Fusion Overfitting by measuring how well encoder representations align across source and target domains. The key intuition is that domain-invariant features should produce similar representations for samples of the same semantic class, regardless of which domain they come from. Conversely, encoders that overfit to source-specific features will produce representations that diverge across domains, since these features are domain dependent.

*Table 1.* Source-target domain representation alignment on EPIC-Kitchens. We measure class-conditional alignment between source and target domain representations using CKA Linear, CKA RBF, and Procrustes similarity. Higher values indicate better preservation of domain-invariant features. For individual encoders, percentages show relative change compared to independently trained unimodal encoders. For the fused representation, percentages show relative change compared to standard Fusion without MER-DG.

| Enc. | Method | CKA-Lin | CKA-RBF | Procrustes |
|---|---|---|---|---|
| Video | Uni-Video | 0.186 | 0.296 | 0.524 |
| | Fusion | 0.147(-21.0%) | 0.232(-21.6%) | 0.422(-19.5%) |
| | Fusion + MER | **0.191**(+2.7%) | **0.324**(+9.5%) | **0.551**(+5.2%) |
| Audio | Uni-Audio | 0.263 | 0.355 | 0.584 |
| | Fusion | 0.187(-28.9%) | 0.289(-18.6%) | 0.503(-13.9%) |
| | Fusion + MER | **0.287**(+9.1%) | **0.381**(+7.3%) | **0.616**(+5.5%) |
| Fused | Fusion | 0.221 | 0.291 | 0.592 |
| | Fusion + MER | **0.311**(+40.7%) | **0.449**(+54.3%) | **0.721**(+21.8%) |

We employ three complementary alignment metrics. Centered Kernel Alignment (CKA) (Kornblith et al., 2019) quantifies representational similarity by comparing kernel matrices; we use both linear and RBF kernels to capture linear and non-linear relationships, respectively. Procrustes similarity (Kornblith et al., 2019) measures how well two sets of representations can be aligned via optimal orthogonal transformation, providing a geometric measure of structural correspondence. For each metric, we compute similarity between source and target features within each class, then average across classes. This class-conditional approach isolates domain-induced differences from semantic content differences: low alignment indicates that the encoder produces domain-dependent representations, while high alignment indicates preservation of domain-invariant features.

Table 1 presents the results. Fusion training substantially degrades cross-domain alignment for both encoders relative to their independently trained counterparts. The Video encoder exhibits alignment reductions of 19–22% across all metrics, while the Audio encoder shows even more pronounced degradation, with CKA-Linear dropping by 28.9%. This consistent degradation across both modalities confirms that the fusion objective causes encoders to overfit to source-specific features. Applying MER-DG restores cross-domain alignment to levels comparable to or exceeding independently trained encoders, with improvements of 21–54% on the fused representation. This recovery demonstrates that entropy regularization counteracts Fusion Overfitting by preserving representational diversity.

**Spectral Analysis.** To complement the alignment analysis, we examine the geometric structure of learned representations through spectral decomposition. If an encoder overfits to a narrow set of source-specific features, its representation will collapse into a low-dimensional subspace. In contrast, an encoder that preserves diverse features should maintain higher-dimensional representations.

We quantify this using singular value decomposition (SVD)

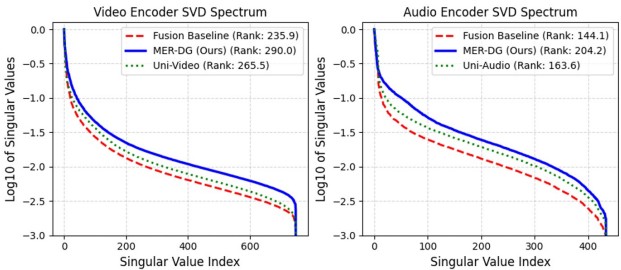

*Figure 2.* Spectral analysis of encoder representations on the EPIC-Kitchens target domain. We plot log-normalized singular values for Video (Left) and Audio (Right) encoders, with RankMe scores in the legend. The Fusion Baseline (red dashed) exhibits rapid singular value decay and lower effective rank, indicating that fusion training compresses representations into low-rank subspaces. MER-DG (blue solid) counteracts this collapse, achieving higher RankMe scores than both the Fusion Baseline and independently trained encoders (green dotted).

of the feature matrix and measure effective rank with the RankMe metric ([Garrido et al., 2023](#)):

$$\text{RankMe}(Z) = \exp\left(-\sum_{i=1}^{d} p_i \log p_i\right), \; p_i = \frac{\sigma_i}{\sum_j \sigma_j}, \tag{3}$$

where $\sigma_i$ denotes the $i$-th singular value of the feature matrix $Z$. RankMe equals the full dimension $d$ when all singular values are equal, indicating maximally diverse representations, and approaches 1 when a single singular value dominates, indicating complete collapse.

Figure 2 plots the log-normalized singular values of feature embeddings on the EPIC-Kitchens target domain, with RankMe scores reported in the legend. Both Video and Audio encoders trained within the Fusion framework exhibit rapid singular value decay, yielding substantially lower RankMe scores (235.9 and 144.1) compared to their independently trained counterparts (265.5 and 163.6). This spectral collapse indicates that the fusion objective causes encoders to discard feature dimensions, compressing representations into low-rank subspaces. Together with the alignment degradation reported in Table 1, these findings establish that fusion training causes encoders to overfit to source-specific cross-modal features at the cost of representational diversity. MER-DG counteracts this collapse, achieving higher RankMe scores than both fusion baselines and independently trained encoders, demonstrating that entropy regularization preserves diversity by keeping all feature dimensions active.

**Domain Classification Analysis.** The alignment and spectral analyses above show that fusion training produces representations that diverge across domains and collapse into low-rank subspaces, while MER-DG reverses both effects. However, these are still indirect measures of domain invari-

*Table 2.* Domain invariance verification on EPIC-Kitchens. We measure effective rank (RankMe) and 3-way domain classification accuracy (Domain Clf Acc) on frozen encoder features. Fusion training reduces RankMe while increasing domain classification accuracy, indicating compression toward domain-specific features. MER-DG simultaneously increases RankMe and decreases domain classification accuracy, confirming that the preserved features are domain-invariant rather than arbitrary.

| Encoder | Method | RankMe | Domain Clf Acc |
|---------|--------|--------|----------------|
| Video | Uni-Video | 265.5 | 81.87 |
| | Fusion | 235.9 | 82.69 |
| | Fusion + MER | **290.0** | **79.58** |
| Audio | Uni-Audio | 163.6 | 65.41 |
| | Fusion | 144.1 | 67.14 |
| | Fusion + MER | **204.2** | **62.59** |

ance: in principle, they could reflect other forms of representation drift rather than a specific bias toward domain-specific features. To directly test whether fusion training shifts representations toward domain-specific features, we train a 3-way domain classifier on frozen encoder features extracted from all three EPIC-Kitchens domains and measure its accuracy in predicting domain identity. Higher domain classification accuracy indicates that encoder features carry more domain-specific information, while lower accuracy indicates greater domain invariance. If MER-DG preserves features indiscriminately, domain classification accuracy should rise alongside effective rank, since a wider variety of retained features would include both domain-invariant and domain-specific signals. If MER-DG preferentially preserves domain-invariant features, domain classification accuracy should decrease even as effective rank increases.

Table 2 reports the results. Fusion training increases domain classification accuracy relative to independently trained encoders for both modalities, confirming that Fusion Overfitting compresses representations toward domain-specific features. With MER-DG, effective rank increases substantially while domain classification accuracy decreases, achieving stronger domain invariance than even independently trained encoders. The simultaneous increase in rank and decrease in domain-specificity indicates that MER-DG preserves features selectively rather than indiscriminately. Together with the alignment and spectral analyses, these results provide direct evidence that fusion training discards domain-invariant features and that MER-DG preserves them.

**Performance Analysis.** To analyze the effect of fusion training on individual encoders, we compare independently trained unimodal models against encoders extracted from jointly-trained multimodal models on the held-out target domain of EPIC-Kitchens.

Figure 3 presents the results. Both encoders trained within the fusion framework exhibit degraded standalone perfor-

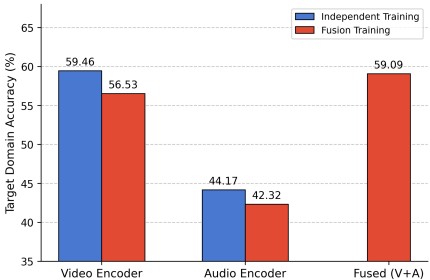

*Figure 3.* Effect of fusion training on encoder performance. Encoders trained within the fusion framework show degraded standalone performance compared to independently trained counterparts. The fused model fails to outperform the independent Video encoder.

mance compared to their independently trained counterparts. The Video encoder drops from 59.46% (independent) to 56.53% (fusion-trained), while the Audio encoder declines from 44.17% to 42.32%. This degradation indicates that fusion training causes encoders to overfit to source-specific features rather than learning domain-invariant ones. Notably, the fused model (59.09%) fails to outperform the independently trained Video encoder (59.46%), despite having access to both modalities. This result confirms that the fusion objective degrades individual encoder performance, directly demonstrating the Fusion Overfitting phenomenon we identified.

**Summary.** Across all four analyses, we observe consistent evidence that fusion training causes encoders to overfit to source-specific cross-modal features: degraded cross-domain alignment, reduced spectral rank, increased domain-specificity, and lower standalone accuracy. These findings motivate our proposed regularization method, which we present in the following section.

## 4. Methodology

To address Fusion Overfitting, we propose Modality-Entropy Regularization for Domain Generalization (MER-DG). Our analysis showed that fusion training causes encoders to collapse into low-rank subspaces, discarding feature dimensions that could encode domain-invariant information. MER-DG counteracts this by maximizing the entropy of each encoder's feature distribution, ensuring that all feature dimensions remain active during training. This preserves feature diversity, enabling robust prediction when source-specific features fail to transfer.

**Feature Entropy Maximization.** We regularize the learning process by maximizing the differential entropy $H(Z)$ of each encoder's feature distribution. We adopt the Log-Determinant entropy estimator from prior work on information maximization (Liu et al., 2022; Erdogan, 2022). However, direct maximization of LD-entropy is numeri-

cally unstable, as entropy can increase through unbounded feature scaling without improving feature diversity. To address this, we decompose the entropy objective into two complementary terms: (i) a marginal-entropy loss that prevents per-dimension collapse, and (ii) a spectral-entropy loss that decorrelates features across dimensions.

Let $Z \in \mathbb{R}^{N \times D}$ be a batch of $N$ embeddings of dimension $D$ from a single encoder. The log-determinant of the covariance matrix provides a principled surrogate for differential entropy (Liu et al., 2022): $H(Z) \propto \log \det \Sigma$, where $\Sigma$ is the covariance matrix. We decompose the covariance matrix as $\Sigma = \Lambda C \Lambda$, where $\Lambda = \text{diag}(\sigma_1, \ldots, \sigma_D)$ contains the standard deviations and $C$ is the correlation matrix.

Substituting this into the LD-entropy definition yields an additive decomposition:

$$\underbrace{\log \det \Sigma}_{\text{LD-Entropy}} = \underbrace{2 \sum_{d=1}^{D} \log \sigma_d}_{\text{Marginal Term}} + \underbrace{\log \det C}_{\text{Spectral Term}}. \quad (4)$$

Equation (4) reveals that maximizing entropy requires two distinct actions: maximizing the marginal spread of each dimension ($\sigma_d$) and minimizing off-diagonal redundancy to maximize the determinant of the correlation matrix. We implement this through two loss terms.

**(1) Marginal-Entropy Loss.** To optimize the first term of Equation (4), we must ensure each dimension maintains sufficient variance. Directly maximizing variance leads to numerical instability, so we instead impose a hinge loss that enforces a variance floor:

$$\mathcal{L}_{\text{marg}}(Z) = \frac{1}{D} \sum_{d=1}^{D} \max\left(0, \gamma - \sigma_d(Z)\right), \quad (5)$$

where $\sigma_d(Z) = \sqrt{\text{Var}(Z_{\cdot,d}) + \epsilon}$ is the standard deviation of the $d$-th feature dimension across the batch, $\epsilon$ is a small constant for numerical stability, and $\gamma$ is a target scale fixed to 1. This guarantees that $\sum \log \sigma_d$ is bounded from below, ensuring every feature dimension remains active.

**(2) Spectral-Entropy Loss.** To optimize the second term, we maximize the log-determinant of the correlation matrix explicitly. Given features $Z$, we first standardize each dimension to zero mean and unit variance, yielding $\hat{Z}$, then compute the empirical correlation matrix $C = \frac{1}{N-1} \hat{Z}^\top \hat{Z}$. The loss is:

$$\mathcal{L}_{\text{spec}}(Z) = -\frac{1}{D} \log \det(C + \epsilon I), \quad (6)$$

where $I \in \mathbb{R}^{D \times D}$ is the identity matrix and $\epsilon$ ensures numerical stability. Since the diagonal of $C$ is fixed at 1,

*Table 3.* **Multi-source DG results on EPIC-Kitchens and HAC.** Results across modalities: **V** (Video), **A** (Audio), and **F** (Flow). We compare four baselines against MER-DG enhanced versions (+MER). **Bold**: best in pair; green: absolute improvement (↑).

| Method | Modality | | | EPIC-Kitchens | | | | HAC | | | |
|---|---|---|---|---|---|---|---|---|---|---|---|
| | V | A | F | D2,D3→D1 | D1,D3→D2 | D1,D2→D3 | Avg | A,C→H | H,C→A | H,A→C | Avg |
| Fusion | ✓ | ✓ | | 53.89 | 63.18 | 60.19 | 59.09 | 70.21 | 69.85 | 51.05 | 63.70 |
| Fusion + MER | ✓ | ✓ | | **58.16** | **67.07** | **63.24** | **62.82**(+3.73) | **83.56** | **78.26** | **52.48** | **71.43**(+7.73) |
| CMC | ✓ | ✓ | | 54.71 | 64.80 | 61.91 | 60.47 | 83.21 | 77.56 | 51.81 | 70.86 |
| CMC + MER | ✓ | ✓ | | **57.93** | **65.67** | **62.83** | **62.14**(+1.67) | **85.36** | **78.48** | **52.91** | **72.25**(+1.39) |
| SimMMDG | ✓ | ✓ | | 57.24 | 65.07 | 63.55 | 61.95 | 72.75 | 76.14 | 54.59 | 67.83 |
| SimMMDG + MER | ✓ | ✓ | | **58.39** | **67.73** | **65.61** | **63.91**(+1.96) | **83.29** | **76.61** | 51.70 | **70.53**(+2.70) |
| CMRF | ✓ | ✓ | | 56.55 | 68.13 | 67.04 | 63.91 | 76.45 | 82.39 | **56.88** | 71.91 |
| CMRF + MER | ✓ | ✓ | | **57.71** | **68.20** | **67.90** | **64.60**(+0.69) | **81.12** | **82.80** | 55.84 | **73.25**(+1.34) |
| Fusion | ✓ | | ✓ | 56.77 | 66.29 | 58.64 | 60.57 | 73.54 | 77.51 | 43.84 | 64.96 |
| Fusion + MER | ✓ | | ✓ | **61.15** | **66.93** | **60.68** | **62.92**(+2.35) | **82.41** | **78.15** | **48.62** | **69.73**(+4.77) |
| CMC | ✓ | | ✓ | 60.46 | 67.60 | 60.37 | 62.81 | 81.26 | 77.70 | 48.17 | 69.05 |
| CMC + MER | ✓ | | ✓ | **61.00** | **68.17** | **60.99** | **63.38**(+0.57) | **84.36** | **78.04** | **50.09** | **70.83**(+1.78) |
| SimMMDG | ✓ | | ✓ | 57.24 | 65.07 | 63.55 | 61.95 | 77.90 | **78.98** | **57.80** | 71.56 |
| SimMMDG + MER | ✓ | | ✓ | **63.68** | **68.40** | **65.09** | **65.72**(+3.77) | **83.71** | 77.17 | 56.67 | **72.51**(+0.95) |
| CMRF | ✓ | | ✓ | **65.28** | 68.13 | 67.04 | 66.82 | 81.16 | 81.25 | **55.50** | 72.64 |
| CMRF + MER | ✓ | | ✓ | 65.18 | **70.12** | **67.55** | **67.62**(+0.80) | **82.07** | **82.81** | 53.27 | **72.72**(+0.08) |
| Fusion | | ✓ | ✓ | 50.59 | 58.65 | 56.91 | 55.38 | 54.45 | 56.91 | 42.16 | 51.17 |
| Fusion + MER | | ✓ | ✓ | **57.47** | **65.47** | **61.70** | **61.55**(+6.17) | **60.71** | **63.02** | **45.50** | **56.41**(+5.24) |
| CMC | | ✓ | ✓ | 54.71 | 64.27 | 61.19 | 60.06 | 61.58 | 62.79 | 45.42 | 56.60 |
| CMC + MER | | ✓ | ✓ | **55.86** | **64.80** | **62.01** | **60.89**(+0.83) | **62.87** | **63.36** | **46.97** | **57.73**(+1.13) |
| SimMMDG | | ✓ | ✓ | 55.86 | 64.60 | 59.34 | 59.93 | 57.88 | 60.79 | **48.62** | 55.76 |
| SimMMDG + MER | | ✓ | ✓ | **56.78** | **69.20** | **64.58** | **63.52**(+3.59) | **62.29** | **64.23** | 45.04 | **57.19**(+1.43) |
| CMRF | | ✓ | ✓ | 57.24 | 64.94 | **66.12** | 62.77 | 59.06 | 61.79 | 55.04 | 58.63 |
| CMRF + MER | | ✓ | ✓ | **58.92** | **65.17** | 65.41 | **63.17**(+0.40) | **63.37** | **66.12** | **56.68** | **62.06**(+3.43) |
| Fusion | ✓ | ✓ | ✓ | 55.21 | 67.95 | 61.25 | 61.47 | 72.33 | 72.18 | 53.66 | 66.06 |
| Fusion + MER | ✓ | ✓ | ✓ | **60.23** | **70.27** | **65.40** | **65.30**(+3.83) | **83.27** | **74.95** | 53.22 | **70.48**(+4.42) |
| CMC | ✓ | ✓ | ✓ | 60.92 | 69.20 | 66.12 | 65.41 | 82.20 | 75.48 | 50.92 | 69.53 |
| CMC + MER | ✓ | ✓ | ✓ | **61.84** | **69.73** | **67.78** | **66.45**(+1.04) | **83.99** | **77.04** | **52.67** | **71.23**(+1.70) |
| SimMMDG | ✓ | ✓ | ✓ | 62.08 | 66.13 | 64.40 | 64.20 | 76.27 | 77.70 | **56.42** | 70.13 |
| SimMMDG + MER | ✓ | ✓ | ✓ | **63.22** | **70.27** | **67.56** | **66.74**(+2.54) | **77.72** | **79.12** | 55.84 | **70.89**(+0.76) |
| CMRF | ✓ | ✓ | ✓ | **61.88** | 70.13 | 70.12 | 67.36 | 78.26 | 79.54 | **60.09** | 72.63 |
| CMRF + MER | ✓ | ✓ | ✓ | 61.84 | **71.73** | **71.55** | **68.37**(+1.01) | **80.46** | **81.37** | 59.30 | **73.71**(+1.08) |

the maximum value of $\det(C)$ is 1, achieved only when $C = I$. Thus, minimizing this loss forces the features to be uncorrelated. This effectively maximizes the spectral entropy of the representation and encourages information to spread across independent axes rather than collapsing into a low-dimensional subspace.

The final regularizer for modality $m$ combines these terms, where $Z^{(m)} \in \mathbb{R}^{N \times D}$ denotes the batch of encoder outputs for modality $m$:

$$\mathcal{L}_{\mathrm{MER}}(Z^{(m)}) = \alpha_{marg}\mathcal{L}_{\mathrm{marg}}(Z^{(m)}) + \alpha_{spec}\mathcal{L}_{\mathrm{spec}}(Z^{(m)}), \quad (7)$$

where $\alpha_{\mathrm{marg}}$ and $\alpha_{\mathrm{spec}}$ are hyperparameters balancing the two regularization terms.

Our proposed regularization is agnostic to the specific architecture or primary learning objective. It serves as an additive term to the base loss of any multimodal framework. The total training objective is:

$$\mathcal{L}_{\mathrm{total}} = \mathcal{L}_{\mathrm{base}} + \lambda \sum_{m=1}^{M} \mathcal{L}_{\mathrm{MER}}(Z^{(m)}), \quad (8)$$

where $\mathcal{L}_{\mathrm{base}}$ represents the original loss of the chosen framework and $\lambda$ is a hyperparameter controlling the regularization strength.

$\mathcal{L}_{\mathrm{MER}}$ is applied independently to each encoder, preserving the base model's fusion mechanism while regularizing the individual representations. This flexibility allows MER-DG to integrate with any multimodal framework. We validate this by integrating our regularizer into four distinct baselines:

- **Classical Fusion:** A standard late-fusion network trained using cross-entropy loss on the fused features.

- **Cross-Modal Contrastive (CMC):** A fusion framework that adds contrastive losses to the standard classification objective to explicitly align modalities in a shared embedding space.

- **State-of-the-Art MMDG:** Specialized frameworks designed for multimodal domain generalization, specifically SimMMDG (Dong et al., 2023) and CMRF (Fan et al., 2024).

*Table 4.* **Single-source domain generalization on EPIC-Kitchens and HAC.** Models are trained on one domain and tested on others. We compare baselines with MER-DG enhancement (+MER). **Bold**: best in pair. Green: absolute improvement (↑).

| | EPIC-Kitchens | | | | | | | HAC | | | | | | |
| | D1 | | D2 | | D3 | | | H | | A | | C | | |
| | D2 | D3 | D1 | D3 | D1 | D2 | *Avg* | A | C | H | C | H | A | *Avg* |
|---|---|---|---|---|---|---|---|---|---|---|---|---|---|---|
| Source: | | | | | | | | | | | | | | |
| **Method** Target: | | | | | | | | | | | | | | |
| Fusion | 56.87 | **56.08** | 48.43 | 59.24 | 56.39 | 55.08 | 55.35 | **66.23** | **46.88** | 70.08 | **45.68** | 60.56 | 62.47 | 58.65 |
| + MER | **56.93** | 55.65 | **53.10** | **63.55** | **58.16** | **55.65** | **57.17**(+1.82) | 66.12 | 46.20 | **75.99** | 45.62 | **66.26** | **62.93** | **60.52**(+1.87) |
| CMC | 58.40 | 57.08 | 51.26 | 63.35 | 56.55 | 56.27 | 57.15 | **65.45** | 43.11 | 70.08 | **46.97** | 69.65 | **63.14** | 59.73 |
| + MER | **59.53** | **57.80** | **52.64** | **64.76** | **57.01** | **57.10** | **58.14**(+0.99) | 65.23 | **44.73** | **77.51** | 46.42 | **69.72** | 62.93 | **61.09**(+1.36) |
| SimMMDG | 54.13 | 57.90 | 50.57 | 63.04 | **60.69** | 64.27 | 58.43 | 64.77 | 39.44 | 71.38 | **50.46** | 60.14 | 70.77 | 59.49 |
| + MER | **59.47** | **57.91** | **54.02** | **63.96** | 58.39 | **67.60** | **60.23**(+1.80) | **67.33** | **41.27** | **76.77** | 49.74 | **67.85** | **71.97** | **62.49**(+3.00) |
| CMRF | 60.80 | 56.78 | 55.17 | 64.99 | 57.24 | 65.73 | 60.12 | 68.75 | **46.33** | 73.55 | 58.26 | 65.22 | 72.46 | 64.10 |
| + MER | **62.13** | **57.39** | **56.02** | **65.71** | **58.85** | **68.80** | **61.48**(+1.36) | **69.21** | 46.11 | **75.95** | **60.21** | **66.85** | **73.28** | **65.27**(+1.17) |

## 5. Experiments

### 5.1. Experimental Setting

**Dataset and Implementation Details.** We evaluate our method on two standard multimodal domain generalization benchmarks: EPIC-Kitchens (Damen et al., 2018) and Human-Animal-Cartoon (HAC) (Dong et al., 2023). Both datasets provide three modalities per sample: RGB video, optical flow, and audio. The EPIC-Kitchens dataset includes three domains (D1, D2, and D3), while HAC comprises three domains representing cartoons (C), humans (H), and animals (A). Our training and evaluation protocol, including domain splits and modality usage, follows the configuration in prior MMDG work (Dong et al., 2023; Fan et al., 2024). Unless otherwise specified, all models use identical backbone architectures and optimization settings. Full implementation details are provided in Appendix A.

**Baselines.** We integrate MER-DG as a plug-in regularizer into four multimodal frameworks and report results with and without regularization: (1) **Fusion**, a standard late-fusion classifier trained with cross-entropy loss; (2) **Cross-Modal Contrastive (CMC)**, which adds supervised contrastive loss to enforce class-consistent alignment across modalities; and (3) **SimMMDG** (Dong et al., 2023) and (4) **CMRF** (Fan et al., 2024), two state-of-the-art methods for multimodal domain generalization. Following standard protocol, we select the checkpoint with best in-domain validation accuracy and report Top-1 accuracy on the held-out target domain.

### 5.2. Results

**Multimodal Multi-source DG.** Table 3 presents results on EPIC-Kitchens and HAC under the multi-source setting, where models are trained on multiple source domains and evaluated on a held-out target domain. We conduct experiments combining any two modalities as well as all three modalities. All experiments are averaged over 3 random seeds; standard deviations are reported in Appendix C. MER-DG consistently improves performance across all baselines and modality combinations, with gains up to

*Table 5.* Comparison of unimodal training with fusion training on EPIC-Kitchens.

| | In-Domain | | | Out-Domain | | |
| Method | V | A | V+A | V | A | V+A |
|---|---|---|---|---|---|---|
| Uni-Video | **76.45** | – | – | 59.46 | – | – |
| Uni-Audio | – | **55.37** | – | – | **44.17** | – |
| Uni-Video + MER | 76.38 | – | – | **59.51** | – | – |
| Uni-Audio + MER | – | 55.34 | – | – | 44.01 | – |
| V+A Fusion | 73.28 | 52.49 | **77.21** | 56.53 | 42.32 | 59.09 |
| V+A Fusion + MER | 75.92 | 54.97 | 76.88 | 58.61 | 43.71 | **62.82** |

7.73%. The Fusion baseline benefits most, achieving an average improvement of 4.8% across all settings. Specialized MMDG methods also improve, with SimMMDG gaining 2.2% and CMRF gaining 1.1% on average. The larger gains on simpler baselines are expected, as specialized methods already partially address representation degradation through their own mechanisms.

**Multimodal Single-source DG.** We further evaluate MER-DG in the more challenging single-source setting, where the model is trained on a single source domain and must generalize to multiple unseen target domains. Table 4 presents results using all three modalities. Consistent with multi-source experiments, MER-DG yields improvements across all baselines on both datasets. On HAC, MER-DG improves the Fusion baseline by 1.87% and SimMMDG by 3.00%. On EPIC-Kitchens, improvements are 1.82% and 1.80% for Fusion and SimMMDG, respectively.

**Uni-modal Performance in MMDG.** A key prediction of our analysis is that fusion training causes encoders to lose domain-invariant features. If MER-DG preserves these features, encoders trained with MER-DG should exhibit improved standalone performance compared to standard fusion training. Table 5 tests this prediction on EPIC-Kitchens.

The table compares independently trained unimodal models against encoders extracted from jointly-trained multimodal models. The top rows show unimodal models trained in isolation: Uni-Video, Uni-Audio. The bottom rows show

*Table 6.* Ablation of marginal and spectral entropy loss.

| $\mathcal{L}_{\mathrm{marg}}$ | $\mathcal{L}_{\mathrm{spec}}$ | D2,D3→D1 | D1,D3→D2 | D1,D2→D3 | Avg. |
|---|---|---|---|---|---|
| − | − | 53.90 | 63.18 | 60.19 | 59.09 |
| ✓ | − | 58.16 | 66.13 | 63.35 | 62.55 |
| − | ✓ | 56.32 | 64.93 | 62.32 | 61.19 |
| ✓ | ✓ | **58.16** | **67.07** | **63.24** | **62.82** |

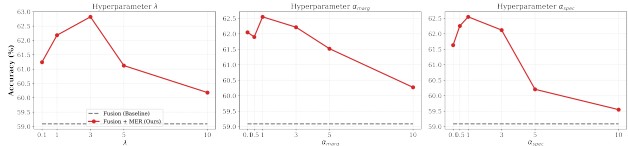

*Figure 4.* **Parameter Sensitivity.** Impact of $\lambda$ (Left), $\alpha_{\mathrm{marg}}$ (Middle), and $\alpha_{\mathrm{spec}}$ (Right) on EPIC-Kitchens. Dashed line: baseline.

the Video+Audio Fusion model along with the performance of its individual encoder branches when detached and evaluated independently. We report both out-of-domain accuracy on the held-out target domain and in-domain accuracy on the source domains. This comparison reveals whether fusion training preserves or degrades each encoder's standalone predictive capacity.

As established in Section 3.2, fusion training degrades standalone encoder performance on the target domain. MER-DG recovers this degradation. The Video encoder improves from 56.53% to 58.61%, and the Audio encoder improves from 42.32% to 43.71%, approaching independently trained levels. The fused model improves from 59.09% to 62.82%. Notably, in-domain performance remains largely unchanged for fusion. MER-DG improves generalization without sacrificing source-domain accuracy. Furthermore, applying MER-DG to independently trained unimodal models yields no significant improvement (Uni-Video: 59.46% vs Uni-Video+MER: 59.51%). This confirms that MER-DG addresses feature loss caused by fusion training rather than providing a general training benefit.

### 5.3. Ablation Studies

**Ablation of Loss Terms.** MER-DG contains two main terms: the marginal-entropy loss $\mathcal{L}_{\mathrm{marg}}$ and the spectral-entropy loss $\mathcal{L}_{\mathrm{spec}}$. We conduct ablation experiments to verify the effectiveness of each term on EPIC-Kitchens with video-audio data.

Table 6 presents the results. Applying only $\mathcal{L}_{\mathrm{marg}}$ improves performance by ensuring a variance floor (62.55%). Similarly, using $\mathcal{L}_{\mathrm{spec}}$ alone provides gains by encouraging decorrelation (61.19%). However, neither term achieves maximal improvement on its own. Combining both terms achieves the best results (62.82%), showing that both are critical for ensuring the representation is both informative and diverse.

**Parameter Sensitivity.** Figure 4 illustrates the impact of varying the global weight $\lambda$ and the component weights $\alpha_{\mathrm{marg}}, \alpha_{\mathrm{spec}}$. We select hyperparameters based on source-domain validation accuracy. For $\lambda$, performance improves steadily, peaking at $\lambda = 3$. While setting $\lambda$ too high ($\geq 5$) leads to a decrease as the regularization interferes with classification, the sensitivity plots demonstrate that our method consistently outperforms the baseline (dashed grey line)

across the entire range of values tested for all three hyperparameters. This indicates that MER-DG is not overly sensitive to precise hyperparameter tuning, provided the weights are within a reasonable range.

## 6. Conclusion

In this paper, we identify Fusion Overfitting, a failure mode in multimodal domain generalization where end-to-end fusion training causes encoders to overfit to source-specific cross-modal co-occurrences rather than learning domain-invariant features. To address this, we propose Modality-Entropy Regularization for Domain Generalization (MER-DG), an architecture-agnostic regularizer that preserves representational diversity across encoder representations. MER-DG integrates as an additive loss term with minimal computational overhead. Extensive experiments on EPIC-Kitchens and HAC benchmarks demonstrate consistent improvements across four baselines: ∼5% on standard fusion and ∼2% over state-of-the-art methods. Crucially, MER-DG also restores standalone encoder performance to near-independent training levels, confirming that entropy regularization directly counteracts the representation collapse induced by fusion training.

## Acknowledgements

This work is partially supported by funding from Amazon Lab126, the Franklin Foundation, and the ML4Seismic Industry Partners at Georgia Tech.

This work used the Delta system at NCSA through an ACCESS Exchange request (project CIS250967), supported by the National Science Foundation under awards OAC-2005572 and ACCESS grants #2138259, #2138286, #2138307, #2137603, and #2138296.

## Impact Statement

This paper presents work whose goal is to advance the field of Machine Learning. There are many potential societal consequences of our work, none which we feel must be specifically highlighted here.

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

# A. Additional Implementation Details

We evaluate our method on two standard multimodal domain generalization benchmarks: EPIC-Kitchens (Damen et al., 2018) and Human-Animal-Cartoon (HAC) (Dong et al., 2023). The EPIC-Kitchens dataset includes eight action classes (*put*, *take*, *open*, *close*, *wash*, *cut*, *mix*, and *pour*) recorded in three kitchens forming domains D1, D2, and D3. The HAC dataset comprises seven action classes (*sleeping*, *watching TV*, *eating*, *drinking*, *swimming*, *running*, and *opening door*) performed by humans (H), animals (A), and cartoon figures (C). Both datasets provide three modalities: RGB video, audio, and optical flow. Our training and evaluation protocol follows (Dong et al., 2023; Fan et al., 2024).

In our framework, we conduct experiments across three modalities: video, audio, and optical flow, adhering to the implementation described in (Dong et al., 2023). We leverage the MMAction2 toolkit (Contributors, 2020) for our experimental setup. To encode visual information, we utilize the SlowFast network (Feichtenhofer et al., 2019), initialized with pre-trained weights on Kinetics-400 (Kay et al., 2017). For the audio encoder, we employ ResNet-18 (He et al., 2016), initialized with weights from the VGGSound pre-trained checkpoint (Chen et al., 2020). The optical flow encoder uses the SlowFast network's slow-only pathway with Kinetics-400 pre-trained weights. The dimensions of the unimodal feature $h$ are 2304 for video, 512 for audio, and 2048 for optical flow. We use the Adam optimizer with a learning rate of 0.0001 and a batch size of 48.

For MER-DG, we set the global regularization weight $\lambda = 3.0$, and the component weights $\alpha_{\mathrm{marg}} = \alpha_{\mathrm{spec}} = 1.0$. The variance floor threshold $\gamma$ in Eq. (5) is set to 1.0, and we use $\epsilon = 10^{-4}$ for numerical stability. When integrating MER-DG with baseline methods (SimMMDG, CMRF), we maintain all original hyperparameters as specified in their respective papers. All experiments were conducted on an NVIDIA A40 GPU. The model is trained for 25 epochs, taking approximately one hour.

# B. Computational Overhead

We analyze the computational cost of MER-DG to demonstrate its practical applicability.

**Complexity Analysis.** Let $N$ denote the batch size and $D$ the feature dimension. The marginal entropy term $\mathcal{L}_{\mathrm{marg}}$ (Eq. (5)) requires $\mathcal{O}(ND)$ operations for per-dimension variance computation. The spectral entropy term $\mathcal{L}_{\mathrm{spec}}$ (Eq. (6)) constructs the $D \times D$ correlation matrix in $\mathcal{O}(ND^2)$ and computes its log-determinant via Cholesky decomposition in $\mathcal{O}(D^3)$, yielding a total complexity of $\mathcal{O}(ND^2 + D^3)$ per modality. In our experiments, $N = 48$ and $D$ ranges from 512 (audio) to 2304 (video); in all cases, both terms are negligible compared to the backbone forward pass.

**Empirical Measurements.** Table 7 reports timing measurements on EPIC-Kitchens using Video+Audio modalities. MER-DG adds 1.2% total training time overhead with negligible memory increase. The regularizer operates only on encoder outputs and requires no additional forward passes through the backbone networks.

*Table 7.* Computational overhead of MER-DG on EPIC-Kitchens (Video+Audio, batch size 48, NVIDIA TITAN RTX).

| Metric | Baseline | + MER-DG |
|---|---|---|
| Per-batch time (ms) | 2034 | 2058 (+1.2%) |
| Peak memory (MB) | 6557 | 6557 |

# C. More Results

*Table 8.* **Multimodal single-source domain generalization results on EPIC-Kitchens and HAC datasets with video and audio modalities.** In this setting, models are trained on a single source domain (indicated in the *Source* row) and evaluated on the remaining unseen target domains (indicated in the *Target* row). We compare four baselines (Standard Fusion, CMC, SimMMDG, and CMRF) against their MER-DG-enhanced versions (+MER). **Bold** values indicate the superior result within each baseline-vs-MER pair, and green numbers denote the absolute improvement (↑) in average accuracy provided by our method.

| | | EPIC-Kitchens | | | | | | | HAC | | | | | | |
| | Source: | D1 | | D2 | | D3 | | | H | | A | | C | | |
| **Method** | Target: | D2 | D3 | D1 | D3 | D1 | D2 | *Avg* | A | C | H | C | H | A | *Avg* |
| Fusion | | 54.93 | **53.49** | 49.66 | **61.29** | 51.03 | **53.49** | 53.98 | **64.00** | **43.88** | 68.08 | **44.68** | 61.56 | 64.47 | 57.78 |
| Fusion + MER | | **57.60** | 52.98 | **51.49** | 60.88 | **51.95** | 52.98 | **54.65**(+0.67) | 63.89 | 43.20 | **73.99** | 44.62 | **67.26** | **64.93** | **59.65**(+1.87) |
| CMC | | 56.12 | 52.28 | 49.20 | 61.12 | 49.46 | 52.46 | 53.44 | **63.22** | 40.11 | 68.08 | **45.97** | 64.65 | **65.14** | 57.86 |
| CMC + MER | | **58.27** | **53.29** | **50.12** | **62.32** | **50.35** | **53.29** | **54.61**(+1.17) | 63.00 | **41.73** | **75.51** | 45.42 | **69.62** | 64.93 | **60.03**(+2.17) |
| SimMMDG | | 53.23 | 53.19 | **48.55** | 60.13 | 50.28 | 59.50 | 54.15 | 63.54 | 42.37 | 71.81 | **47.43** | 66.95 | 68.32 | 60.07 |
| SimMMDG + MER | | **53.87** | **54.42** | 47.13 | **60.47** | **54.71** | **62.73** | **55.56**(+1.41) | **66.10** | **44.20** | **77.20** | 46.71 | **72.67** | **69.52** | **62.73**(+2.66) |
| CMRF | | 54.62 | 53.49 | 53.57 | 59.15 | 50.57 | 62.06 | 55.58 | 67.22 | **44.70** | 74.93 | 49.82 | 70.09 | 70.82 | 62.93 |
| CMRF + MER | | **57.60** | **55.03** | **54.26** | **64.58** | **52.95** | **63.87** | **58.05**(+2.47) | **67.68** | 44.48 | **77.33** | **51.77** | **71.72** | **71.64** | **64.10**(+1.17) |

*Table 9.* **Multimodal single-source domain generalization results on EPIC-Kitchens and HAC datasets with video and flow modalities.** In this setting, models are trained on a single source domain (indicated in the *Source* row) and evaluated on the remaining unseen target domains (indicated in the *Target* row). We compare four baselines (Standard Fusion, CMC, SimMMDG, and CMRF) against their MER-DG-enhanced versions (+MER). **Bold** values indicate the superior result within each baseline-vs-MER pair, and green numbers denote the absolute improvement (↑) in average accuracy provided by our method.

| | | EPIC-Kitchens | | | | | | | HAC | | | | | | |
| | Source: | D1 | | D2 | | D3 | | | H | | A | | C | | |
| **Method** | Target: | D2 | D3 | D1 | D3 | D1 | D2 | *Avg* | A | C | H | C | H | A | *Avg* |
| Fusion | | **54.80** | 51.03 | 51.49 | **60.06** | 54.94 | 51.03 | 53.89 | **64.52** | **43.88** | 71.08 | **42.68** | 59.56 | 58.47 | 56.70 |
| Fusion + MER | | 54.20 | **51.97** | **54.02** | 59.86 | **55.17** | **51.97** | **54.53**(+0.64) | 64.42 | 43.20 | **75.99** | 42.62 | **63.32** | **58.93** | **58.08**(+1.38) |
| CMC | | 53.20 | 51.28 | 52.64 | 59.96 | 54.33 | 49.28 | 53.45 | **63.95** | 40.11 | 72.08 | **43.97** | 65.65 | **59.14** | 57.48 |
| CMC + MER | | **55.42** | **52.51** | **53.99** | **61.16** | **55.13** | **50.28** | **54.75**(+1.30) | 63.73 | **41.73** | **78.51** | 43.42 | **67.72** | 58.93 | **59.01**(+1.53) |
| SimMMDG | | 55.84 | 52.69 | 52.21 | 61.85 | 56.86 | 59.12 | 56.43 | 65.78 | 41.76 | 75.29 | **48.10** | 65.61 | 56.86 | 58.90 |
| SimMMDG + MER | | **56.87** | **54.42** | **54.13** | **62.47** | **57.71** | **61.15** | **57.79**(+1.36) | **68.34** | **43.18** | **80.68** | 47.38 | **67.23** | **58.06** | **60.81**(+1.91) |
| CMRF | | **57.02** | 53.13 | 56.08 | 61.37 | **58.61** | 61.69 | 57.98 | 67.07 | **44.27** | 74.21 | 50.55 | 69.31 | 62.46 | 61.31 |
| CMRF + MER | | 56.93 | **57.67** | **58.09** | **63.45** | 57.24 | **62.13** | **59.25**(+1.27) | **67.53** | 44.05 | **76.61** | **52.50** | **70.94** | **63.28** | **62.49**(+1.18) |

*Table 10.* **Multimodal single-source domain generalization results on EPIC-Kitchens and HAC datasets with audio and flow modalities.** In this setting, models are trained on a single source domain (indicated in the *Source* row) and evaluated on the remaining unseen target domains (indicated in the *Target* row). We compare four baselines (Standard Fusion, CMC, SimMMDG, and CMRF) against their MER-DG-enhanced versions (+MER). **Bold** values indicate the superior result within each baseline-vs-MER pair, and green numbers denote the absolute improvement (↑) in average accuracy provided by our method.

| | | EPIC-Kitchens | | | | | | | HAC | | | | | | |
| | Source: | D1 | | D2 | | D3 | | | H | | A | | C | | |
| Method | Target: | D2 | D3 | D1 | D3 | D1 | D2 | *Avg* | A | C | H | C | H | A | *Avg* |
|---|---|---|---|---|---|---|---|---|---|---|---|---|---|---|---|
| Fusion | | 53.60 | **55.44** | 44.60 | **59.34** | 50.58 | **55.44** | 53.17 | **58.23** | **36.88** | 53.08 | **38.68** | 36.56 | 43.47 | 44.48 |
| Fusion + MER | | **55.07** | 54.52 | **49.89** | 58.70 | **54.48** | 54.52 | **54.53**(+1.36) | 58.12 | 36.20 | **58.99** | 38.62 | **42.26** | **43.93** | **46.35**(+1.87) |
| CMC | | 54.15 | 54.10 | 45.10 | 59.10 | 51.10 | 52.14 | 52.62 | **57.45** | 33.11 | 53.08 | **39.97** | 45.65 | **44.14** | 45.57 |
| CMC + MER | | **55.47** | **54.49** | **47.13** | **59.96** | **53.33** | **53.49** | **53.98**(+1.36) | 57.23 | **34.73** | **60.51** | 39.42 | **45.72** | 43.93 | **46.92**(+1.35) |
| SimMMDG | | 54.63 | 55.13 | **48.52** | 60.40 | 54.03 | 62.44 | 55.86 | 59.16 | 37.68 | 56.04 | **40.62** | 38.81 | 45.84 | 46.36 |
| SimMMDG + MER | | **55.87** | **56.42** | 47.13 | **61.47** | **54.71** | **66.73** | **57.06**(+1.20) | **61.72** | **39.51** | **61.43** | 39.90 | **46.52** | **47.04** | **49.35**(+2.99) |
| CMRF | | 54.06 | 55.21 | 49.06 | 61.46 | 52.92 | 62.68 | 55.90 | 59.60 | **37.90** | 59.89 | 44.06 | 42.85 | 48.09 | 48.73 |
| CMRF + MER | | **57.07** | **56.13** | **50.43** | **62.68** | **53.03** | **63.53** | **57.15**(+1.25) | **60.06** | 37.68 | **62.29** | **46.01** | **44.48** | **48.91** | **49.91**(+1.18) |

*Table 11.* **Multi-source domain generalization results on EPIC-Kitchens and HAC benchmarks with standard deviations.** We evaluate four multimodal fusion baselines (Fusion, CMC, SimMMDG, and CMRF) and their MER-DG enhanced counterparts (+MER) across all modality combinations: Video+Audio (V+A), Video+Flow (V+F), Audio+Flow (A+F), and all three modalities (V+A+F). All results are averaged over 3 random seeds, with standard deviation reported in subscript (±std) to demonstrate statistical significance of the improvements. **Bold** indicates the best performance within each baseline/+MER pair.

| | Modality | | | EPIC-Kitchens | | | | HAC | | | |
| Method | V | A | F | D2,D3→D1 | D1,D3→D2 | D1,D2→D3 | Avg | A,C→H | H,C→A | H,A→C | Avg |
|---|---|---|---|---|---|---|---|---|---|---|---|
| Fusion | ✓ | ✓ | | $53.89_{\pm0.28}$ | $63.18_{\pm0.22}$ | $60.19_{\pm0.31}$ | 59.09 | $70.21_{\pm0.32}$ | $69.85_{\pm0.35}$ | $51.05_{\pm0.38}$ | 63.70 |
| Fusion + MER | ✓ | ✓ | | $\mathbf{58.16_{\pm0.25}}$ | $\mathbf{67.07_{\pm0.19}}$ | $\mathbf{63.24_{\pm0.28}}$ | **62.82** | $\mathbf{83.56_{\pm0.27}}$ | $\mathbf{78.26_{\pm0.31}}$ | $\mathbf{52.48_{\pm0.34}}$ | **71.43** |
| CMC | ✓ | ✓ | | $54.71_{\pm0.24}$ | $64.80_{\pm0.20}$ | $61.91_{\pm0.27}$ | 60.47 | $83.21_{\pm0.22}$ | $77.56_{\pm0.26}$ | $51.81_{\pm0.32}$ | 70.86 |
| CMC + MER | ✓ | ✓ | | $\mathbf{57.93_{\pm0.21}}$ | $\mathbf{65.67_{\pm0.18}}$ | $\mathbf{62.83_{\pm0.24}}$ | **62.14** | $\mathbf{85.36_{\pm0.19}}$ | $\mathbf{78.48_{\pm0.23}}$ | $\mathbf{52.91_{\pm0.29}}$ | **72.25** |
| SimMMDG | ✓ | ✓ | | $57.24_{\pm0.29}$ | $65.07_{\pm0.24}$ | $63.55_{\pm0.22}$ | 61.95 | $72.75_{\pm0.33}$ | $76.14_{\pm0.28}$ | $54.59_{\pm0.36}$ | 67.83 |
| SimMMDG + MER | ✓ | ✓ | | $\mathbf{58.39_{\pm0.26}}$ | $\mathbf{67.73_{\pm0.21}}$ | $\mathbf{65.61_{\pm0.19}}$ | **63.91** | $\mathbf{83.29_{\pm0.28}}$ | $\mathbf{76.61_{\pm0.25}}$ | $51.70_{\pm0.32}$ | **70.53** |
| CMRF | ✓ | ✓ | | $56.55_{\pm0.20}$ | $68.13_{\pm0.17}$ | $67.04_{\pm0.21}$ | 63.91 | $76.45_{\pm0.26}$ | $82.39_{\pm0.23}$ | $\mathbf{56.88_{\pm0.30}}$ | 71.91 |
| CMRF + MER | ✓ | ✓ | | $\mathbf{57.71_{\pm0.18}}$ | $\mathbf{68.20_{\pm0.15}}$ | $\mathbf{67.90_{\pm0.19}}$ | **64.60** | $\mathbf{81.12_{\pm0.23}}$ | $\mathbf{82.80_{\pm0.21}}$ | $55.84_{\pm0.27}$ | **73.25** |
| Fusion | ✓ | | ✓ | $56.77_{\pm0.29}$ | $66.29_{\pm0.24}$ | $58.64_{\pm0.33}$ | 60.57 | $73.54_{\pm0.31}$ | $77.51_{\pm0.27}$ | $43.84_{\pm0.39}$ | 64.96 |
| Fusion + MER | ✓ | | ✓ | $\mathbf{61.15_{\pm0.26}}$ | $\mathbf{66.93_{\pm0.21}}$ | $\mathbf{60.68_{\pm0.29}}$ | **62.92** | $\mathbf{82.41_{\pm0.27}}$ | $\mathbf{78.15_{\pm0.24}}$ | $\mathbf{48.62_{\pm0.34}}$ | **69.73** |
| CMC | ✓ | | ✓ | $60.46_{\pm0.24}$ | $67.60_{\pm0.20}$ | $60.37_{\pm0.27}$ | 62.81 | $81.26_{\pm0.22}$ | $77.70_{\pm0.24}$ | $48.17_{\pm0.32}$ | 69.05 |
| CMC + MER | ✓ | | ✓ | $\mathbf{61.00_{\pm0.21}}$ | $\mathbf{68.17_{\pm0.18}}$ | $\mathbf{60.99_{\pm0.24}}$ | **63.38** | $\mathbf{84.36_{\pm0.19}}$ | $\mathbf{78.04_{\pm0.21}}$ | $\mathbf{50.09_{\pm0.29}}$ | **70.83** |
| SimMMDG | ✓ | | ✓ | $57.24_{\pm0.30}$ | $65.07_{\pm0.26}$ | $63.55_{\pm0.24}$ | 61.95 | $77.90_{\pm0.28}$ | $78.98_{\pm0.25}$ | $\mathbf{57.80_{\pm0.34}}$ | 71.56 |
| SimMMDG + MER | ✓ | | ✓ | $\mathbf{63.68_{\pm0.27}}$ | $\mathbf{68.40_{\pm0.23}}$ | $\mathbf{65.09_{\pm0.21}}$ | **65.72** | $\mathbf{83.71_{\pm0.25}}$ | $77.17_{\pm0.22}$ | $56.67_{\pm0.30}$ | **72.51** |
| CMRF | ✓ | | ✓ | $\mathbf{65.28_{\pm0.20}}$ | $68.13_{\pm0.17}$ | $67.04_{\pm0.20}$ | 66.82 | $81.16_{\pm0.22}$ | $81.25_{\pm0.24}$ | $\mathbf{55.50_{\pm0.28}}$ | 72.64 |
| CMRF + MER | ✓ | | ✓ | $65.18_{\pm0.17}$ | $\mathbf{70.12_{\pm0.15}}$ | $\mathbf{67.55_{\pm0.17}}$ | **67.62** | $\mathbf{82.07_{\pm0.19}}$ | $\mathbf{82.81_{\pm0.21}}$ | $53.27_{\pm0.25}$ | **72.72** |
| Fusion | | ✓ | ✓ | $50.59_{\pm0.33}$ | $58.65_{\pm0.30}$ | $56.91_{\pm0.35}$ | 55.38 | $54.45_{\pm0.39}$ | $56.91_{\pm0.36}$ | $42.16_{\pm0.42}$ | 51.17 |
| Fusion + MER | | ✓ | ✓ | $\mathbf{57.47_{\pm0.29}}$ | $\mathbf{65.47_{\pm0.26}}$ | $\mathbf{61.70_{\pm0.31}}$ | **61.55** | $\mathbf{60.71_{\pm0.34}}$ | $\mathbf{63.02_{\pm0.32}}$ | $\mathbf{45.50_{\pm0.38}}$ | **56.41** |
| CMC | | ✓ | ✓ | $54.71_{\pm0.27}$ | $64.27_{\pm0.24}$ | $61.19_{\pm0.28}$ | 60.06 | $61.58_{\pm0.30}$ | $62.79_{\pm0.28}$ | $45.42_{\pm0.34}$ | 56.60 |
| CMC + MER | | ✓ | ✓ | $\mathbf{55.86_{\pm0.24}}$ | $\mathbf{64.80_{\pm0.21}}$ | $\mathbf{62.01_{\pm0.25}}$ | **60.89** | $\mathbf{62.87_{\pm0.27}}$ | $\mathbf{63.36_{\pm0.25}}$ | $\mathbf{46.97_{\pm0.31}}$ | **57.73** |
| SimMMDG | | ✓ | ✓ | $55.86_{\pm0.30}$ | $64.60_{\pm0.27}$ | $59.34_{\pm0.32}$ | 59.93 | $57.88_{\pm0.35}$ | $60.79_{\pm0.32}$ | $\mathbf{48.62_{\pm0.38}}$ | 55.76 |
| SimMMDG + MER | | ✓ | ✓ | $\mathbf{56.78_{\pm0.27}}$ | $\mathbf{69.20_{\pm0.24}}$ | $\mathbf{64.58_{\pm0.28}}$ | **63.52** | $\mathbf{62.29_{\pm0.31}}$ | $\mathbf{64.23_{\pm0.29}}$ | $45.04_{\pm0.34}$ | **57.19** |
| CMRF | | ✓ | ✓ | $57.24_{\pm0.22}$ | $64.94_{\pm0.19}$ | $\mathbf{66.12_{\pm0.17}}$ | 62.77 | $59.06_{\pm0.28}$ | $61.79_{\pm0.26}$ | $55.04_{\pm0.32}$ | 58.63 |
| CMRF + MER | | ✓ | ✓ | $\mathbf{58.92_{\pm0.19}}$ | $\mathbf{65.17_{\pm0.17}}$ | $65.41_{\pm0.15}$ | **63.17** | $\mathbf{63.37_{\pm0.25}}$ | $\mathbf{66.12_{\pm0.23}}$ | $\mathbf{56.68_{\pm0.29}}$ | **62.06** |
| Fusion | ✓ | ✓ | ✓ | $55.21_{\pm0.30}$ | $67.95_{\pm0.26}$ | $61.25_{\pm0.33}$ | 61.47 | $72.33_{\pm0.32}$ | $72.18_{\pm0.35}$ | $53.66_{\pm0.39}$ | 66.06 |
| Fusion + MER | ✓ | ✓ | ✓ | $\mathbf{60.23_{\pm0.27}}$ | $\mathbf{70.27_{\pm0.23}}$ | $\mathbf{65.40_{\pm0.29}}$ | **65.30** | $\mathbf{83.27_{\pm0.28}}$ | $\mathbf{74.95_{\pm0.31}}$ | $53.22_{\pm0.35}$ | **70.48** |
| CMC | ✓ | ✓ | ✓ | $60.92_{\pm0.24}$ | $69.20_{\pm0.20}$ | $66.12_{\pm0.26}$ | 65.41 | $82.20_{\pm0.24}$ | $75.48_{\pm0.28}$ | $50.92_{\pm0.33}$ | 69.53 |
| CMC + MER | ✓ | ✓ | ✓ | $\mathbf{61.84_{\pm0.21}}$ | $\mathbf{69.73_{\pm0.17}}$ | $\mathbf{67.78_{\pm0.23}}$ | **66.45** | $\mathbf{83.99_{\pm0.21}}$ | $\mathbf{77.04_{\pm0.25}}$ | $\mathbf{52.67_{\pm0.29}}$ | **71.23** |
| SimMMDG | ✓ | ✓ | ✓ | $62.08_{\pm0.28}$ | $66.13_{\pm0.26}$ | $64.40_{\pm0.24}$ | 64.20 | $76.27_{\pm0.30}$ | $77.70_{\pm0.28}$ | $\mathbf{56.42_{\pm0.35}}$ | 70.13 |
| SimMMDG + MER | ✓ | ✓ | ✓ | $\mathbf{63.22_{\pm0.25}}$ | $\mathbf{70.27_{\pm0.23}}$ | $\mathbf{67.56_{\pm0.21}}$ | **66.74** | $\mathbf{77.72_{\pm0.27}}$ | $\mathbf{79.12_{\pm0.25}}$ | $55.84_{\pm0.31}$ | **70.89** |
| CMRF | ✓ | ✓ | ✓ | $\mathbf{61.88_{\pm0.20}}$ | $70.13_{\pm0.17}$ | $70.12_{\pm0.20}$ | 67.36 | $78.26_{\pm0.24}$ | $79.54_{\pm0.22}$ | $\mathbf{60.09_{\pm0.28}}$ | 72.63 |
| CMRF + MER | ✓ | ✓ | ✓ | $61.84_{\pm0.17}$ | $\mathbf{71.73_{\pm0.15}}$ | $\mathbf{71.55_{\pm0.17}}$ | **68.37** | $\mathbf{80.46_{\pm0.21}}$ | $\mathbf{81.37_{\pm0.19}}$ | $59.30_{\pm0.25}$ | **73.71** |

# D. Additional Ablation Studies

This appendix presents three additional ablation experiments that probe the boundaries of MER-DG's applicability. We examine the fusion architecture, compare MER-DG against standard regularization strategies, and evaluate robustness under input corruption. All experiments use EPIC-Kitchens with video and audio modalities under the multi-source setting.

## D.1. Fusion Architecture

The main experiments use late fusion to combine encoder outputs. To verify that MER-DG's benefit is not specific to this fusion strategy, we replace the late-fusion module with a cross-attention fusion module while keeping the encoders, training protocol, and dataset fixed. MER-DG operates on encoder outputs prior to fusion and is therefore agnostic to how those outputs are combined, so the regularizer should remain effective under alternative fusion mechanisms. Table 12 reports the results. MER-DG improves cross-attention fusion from 59.48% to 61.13%, consistent with the gains observed under late fusion, confirming that MER-DG addresses fusion-induced feature loss at the encoder level rather than at the fusion module.

*Table 12.* Fusion architecture ablation on EPIC-Kitchens (Video + Audio). MER-DG improves performance under both late fusion and cross-attention fusion, demonstrating that the regularizer is agnostic to the fusion mechanism.

| Fusion Architecture | Baseline | + MER-DG |
|---|---|---|
| Late Fusion | 59.09 | **62.82** |
| Cross-Attention | 59.48 | **61.13** |

## D.2. Comparison with Standard Regularization

A natural question is whether MER-DG's gains stem from its specific design or from a more general regularization effect. To answer this, we compare MER-DG against four standard regularization techniques applied per-encoder to match MER-DG's scope: dropout, Gaussian noise injection on encoder outputs, weight decay, and label smoothing on the classification head. All methods are tuned on source-domain validation accuracy and applied to the standard fusion baseline. Table 13 reports the results. Standard regularization techniques produce only marginal improvements over the baseline, with the strongest method (label smoothing) gaining 1.12%. MER-DG improves the same baseline by 3.73%, more than three times the gain of any standard regularizer. This indicates that MER-DG's benefit derives from its specific design for Fusion Overfitting rather than from generic regularization effects, since methods that successfully regularize encoder representations in other contexts do not produce comparable gains in the multimodal domain generalization setting.

*Table 13.* Comparison with standard regularization on EPIC-Kitchens (Video + Audio). All methods are applied per-encoder to the standard fusion baseline. MER-DG provides substantially larger gains than any standard regularization technique.

| Method | Avg. Accuracy |
|---|---|
| Fusion | 59.09 |
| + Dropout | 59.45 |
| + Noise Injection | 58.96 |
| + Weight Decay | 59.82 |
| + Label Smoothing | 60.21 |
| + MER-DG | **62.82** |

## D.3. Robustness to Input Corruption

The standalone encoder analysis in Section 5.2 shows that MER-DG reduces co-dependence between modalities by restoring each encoder's standalone predictive capacity. To test whether this translates into robustness of the full multimodal model under degraded inputs, we evaluate the trained models on EPIC-Kitchens with corruptions applied at test time. We inject Gaussian noise into one modality's encoder features at two intensities ($\sigma = 0.5$ and $\sigma = 1.0$) and simulate missing modalities by zeroing one encoder's output. The full fusion pipeline, including the fusion head and classifier, is then evaluated on the corrupted inputs.

Table 14 reports the results. MER-DG shows smaller accuracy degradation than the baseline under every corruption condition, and the gap widens under stronger corruption. Under $\sigma = 1.0$ noise on video, the baseline drops by 7.95% while MER-DG drops by 6.55%; under audio drop, the baseline drops by 10.94% while MER-DG drops by 8.55%. The widening

gap indicates that the standalone encoder improvements documented in Table 5 translate into practical robustness of the multimodal system under partial or degraded inputs, which is a common scenario in real-world multimodal deployment.

*Table 14.* Robustness to input corruption on EPIC-Kitchens (Video + Audio). We report accuracy under Gaussian noise on encoder features ($\sigma = 0.5$ and $\sigma = 1.0$) and under modality drop. Drop columns report absolute accuracy reduction from the clean condition. MER-DG shows smaller degradation across all corruption conditions.

| Condition | Fusion | Fusion + MER | Drop (Fusion) | Drop (MER) |
|---|---|---|---|---|
| Clean | 59.09 | 62.82 | – | – |
| Noise Video, $\sigma = 0.5$ | 55.72 | 60.04 | $-3.37$ | $-2.78$ |
| Noise Video, $\sigma = 1.0$ | 51.14 | 56.27 | $-7.95$ | $-6.55$ |
| Noise Audio, $\sigma = 0.5$ | 57.18 | 61.64 | $-1.91$ | $-1.18$ |
| Noise Audio, $\sigma = 1.0$ | 54.73 | 59.86 | $-4.36$ | $-2.96$ |
| Drop Video | 38.47 | 43.18 | $-20.62$ | $-19.64$ |
| Drop Audio | 48.15 | 54.27 | $-10.94$ | $-8.55$ |

### D.4. Component Ablation on Specialized Baselines

The loss term ablation in Section 5.2 examines the contribution of $\mathcal{L}_{\mathrm{marg}}$ and $\mathcal{L}_{\mathrm{spec}}$ on the standard fusion baseline. The standard fusion baseline isolates each component's contribution without confounding effects from other regularization mechanisms. To verify that both components remain complementary when integrated with specialized MMDG methods, we extend the ablation to SimMMDG and CMRF on EPIC-Kitchens with video and audio modalities. Table 15 reports the results. Both loss terms contribute independently to performance on each specialized baseline, and the full MER-DG configuration achieves the best results in both cases. This pattern is consistent with the loss term ablation in the main paper and confirms that the marginal-entropy and spectral-entropy components remain complementary across frameworks rather than being specific to the standard fusion setting.

*Table 15.* Component ablation on specialized baselines. We report accuracy on EPIC-Kitchens (Video + Audio) with each loss term applied individually and in combination. Both $\mathcal{L}_{\mathrm{marg}}$ and $\mathcal{L}_{\mathrm{spec}}$ contribute independently, and the full MER-DG configuration achieves the best results on both SimMMDG and CMRF.

| Method | Baseline | $\mathcal{L}_{\mathrm{spec}}$ only | $\mathcal{L}_{\mathrm{marg}}$ only | Full MER |
|---|---|---|---|---|
| SimMMDG | 61.95 | 63.16 | 62.58 | **63.91** |
| CMRF | 63.91 | 64.25 | 64.12 | **64.60** |

