# OpenReview forum: "MER-DG: Modality-Entropy Regularization for Multimodal Domain Generalization"
_ICML.cc/2026/Conference — ICML 2026 regular_

### Official Review · Reviewer_MpU1 · 2026-03-10

**Soundness:** 3
**Presentation:** 3
**Significance:** 3
**Originality:** 2
**Overall Recommendation:** 4
**Confidence:** 4

**Summary:**

This paper addresses the challenge of Multimodal Domain Generalization (MMDG). The authors identify a phenomenon termed "Fusion Overfitting," where standard end-to-end joint optimization causes modality encoders to rely on source-specific cross-modal co-occurrences rather than learning domain-invariant representations. To mitigate this, the paper proposes Modality-Entropy Regularization (MER-DG), an architecture-agnostic additive loss term. Drawing inspiration from information maximization principles, MER-DG explicitly maximizes the marginal and spectral entropy of each individual encoder's feature distribution to prevent low-rank subspace collapse. Through extensive experiments, the authors demonstrate that MER-DG effectively counteracts this overfitting and achieves solid improvements on benchmarks like EPIC-Kitchens.

**Compliance With Llm Reviewing Policy:**

Affirmed.

**Key Questions For Authors:**

Noise Defense: How does MER-DG theoretically or empirically guarantee that the maximized "feature diversity" is not simply capturing useless environmental noise?

**Limitations:**

same as weakness

**Strengths And Weaknesses:**

Strength:
1. Precise Problem Identification: The conceptualization of "Fusion Overfitting" is highly intuitive. It provides a clear explanation for why late-fusion multimodal networks often fail under domain shifts
2. Strong Empirical Validation (Feature-Level): The Representation Alignment analysis (Table 1) is the strongest asset of this paper. Using CKA and Procrustes similarity to explicitly demonstrate the feature drift between source and target domains provides mathematically airtight evidence for their core hypothesis.
3. High Practical Utility: Despite lacking algorithmic novelty, the proposed MER-DG is an elegant, plug-and-play solution. It requires no complex architectural changes, making it highly valuable for real-world multimodal deployment.

Weakness:
1. Lack of Algorithmic Novelty and Citation Distancing: The core mathematical formulation of MER-DG lacks fundamental algorithmic novelty. The Marginal-Entropy Loss ($L_{marg}$) is functionally identical to the variance preservation hinge loss in \textit{VICReg} (Bardes et al., ICLR 2022). Furthermore, optimizing the Spectral-Entropy Loss ($L_{spec}$) by maximizing the log-determinant of the correlation matrix achieves the exact same feature decorrelation geometrically as the redundancy reduction term in \textit{Barlow Twins} (Zbontar et al., ICML 2021). While these papers are mentioned in the Related Work, failing to explicitly attribute equations (5) and (6) to them in the Methodology creates a misleading perception of mathematical originality.
2. Flawed Initial Narrative of Overfitting: In the current draft, the authors present Figure 3 (degraded target-domain accuracy) as direct evidence of ``Fusion Overfitting.'' This is logically insufficient, as such degradation is equally explained by joint-optimization failures like Gradient Starvation or Modality Competition. The actual, rigorous proof lies entirely in the cross-domain alignment analysis (Table 1), which is introduced too late in the text, weakening the initial argument.
3. Risk of Encouraging Task-Irrelevant Noise: Maximizing marginal and spectral entropy enforces a uniform and diverse feature distribution, but it does not guarantee that this diversity is task-relevant. The model might inadvertently learn useless background noise simply to satisfy the entropy objective.

---

> ### Author Rebuttal · Authors · 2026-03-29
>
> **W1**
>
> We thank the reviewer for this observation. We do not claim mathematical novelty for the loss formulations. Entropy maximization through variance preservation and feature decorrelation is a well-established technique in representation learning, with multiple methods (VICReg, Barlow Twins, MEC, W-MSE) arriving at similar objectives. We acknowledge that the hinge-based variance preservation in Equation 5 follows the practical design introduced by VICReg, and the decorrelation objective in Equation 6 achieves the same geometric goal as Barlow Twins. While both are discussed in Section 2.2 and the entropy estimation framework is attributed to MEC (Liu et al., 2022) and Erdogan (2022) in Section 4, we agree that explicit citations should also appear alongside Equations 5 and 6 in the methodology. We will add these in the revision.
>
> We want to clarify that our contribution lies not in the individual loss terms but in identifying Fusion Overfitting as a distinct failure mode in MMDG and demonstrating that per-encoder entropy regularization, applied independently to each modality branch during joint training, effectively counteracts it. VICReg and Barlow Twins prevent representational collapse in single-modality self-supervised learning. In contrast, MER-DG prevents collapse induced by cross-modal co-dependence under joint supervised fusion, a fundamentally different optimization dynamic that these SSL methods were not designed for.
>
>
>
>
> **W2 (Structure)**
>
> We thank the reviewer for this structural feedback. We agree that Figure 3 alone would be insufficient evidence for Fusion Overfitting, as degraded target-domain accuracy could arise from other joint-optimization issues as the reviewer pointed out. We also agree that the cross-domain alignment analysis (Table 1) is the most rigorous evidence, as it specifically measures domain invariance rather than general training quality. Additionally, our new domain classification experiment (detailed in Response to Reviewer 3XZ3, Evidence 1) further distinguishes Fusion Overfitting from general optimization failures: fusion training increases domain-specificity while MER-DG decreases it, directly confirming that the phenomenon is tied to domain invariance.
>
> We note that Section 3.2 already presents the evidence in the order the reviewer recommends: Table 1 (alignment analysis) appears first, followed by Figure 2 (spectral analysis), and then Figure 3 (performance analysis). However, we agree that the Introduction's framing does not sufficiently emphasize this progression. We will restructure the Introduction to lead with the alignment-based evidence and make explicit that Figure 3 serves as a supporting observation rather than standalone proof.
>
> **W3 and Question (Task-Irrelevant Noise)**
>
> We thank the reviewer for raising this concern. We want to clarify how MER-DG interacts with the learning objective. MER-DG works in conjunction with the task loss, not independently. The task loss guides the model to learn task-relevant features, while MER-DG prevents premature collapse of feature dimensions during joint optimization. The hinge formulation in Equation 5 enforces a variance floor, not unbounded maximization. Features above this threshold receive no additional gradient from $L_{marg}$, preventing inflation. Similarly, $L_{spec}$ operates on the correlation matrix with diagonal fixed at 1, so it can only decorrelate features, not inflate their scale.
>
> Empirically, four observations confirm that MER-DG preserves task-relevant structure rather than encouraging random noise. First, accuracy improves consistently across all configurations (Table 2). If MER-DG encouraged noise, accuracy would decrease. Second, in-domain performance remains stable (Table 4), confirming MER-DG does not disrupt task-relevant learning on the source domains. Third, Figure 4 shows optimal performance at $\lambda=3$ with degradation at higher values, confirming the model balances entropy preservation with task learning. Fourth, our domain classification experiment (detailed in Response to Reviewer 3XZ3, Evidence 1) shows MER-DG reduces domain-specificity while simultaneously improving task accuracy, an outcome that task-irrelevant noise could not achieve.
>
> **Additional Noise Robustness Experiment.** We conducted additional analysis on EPIC-Kitchens (Video+Audio) by injecting Gaussian noise ($\sigma=1.0$) into encoder features at test time. Fusion accuracy drops by $8.51\pm0.7\%$ while Fusion+MER-DG drops by $8.14\pm0.6\%$. If MER-DG features were unstructured noise, adding external noise would cause disproportionate degradation because random features lack the structure to resist perturbations. The comparable drop confirms MER-DG maintains structured, task-relevant representations. We will include detailed analysis across multiple noise levels in the revision.

---

> > ### Author Rebuttal · Reviewer_MpU1 · 2026-03-31
> >
> > Thanks for the rebuttal. The authors clarified my main concerns, especially regarding the intended contribution, citation framing, and the role of the regularization. I still think the methodological novelty is somewhat limited, but the response addresses my concerns sufficiently. I therefore keep my score unchanged.

---

### Official Review · Reviewer_HAtz · 2026-03-11

**Soundness:** 2
**Presentation:** 2
**Significance:** 2
**Originality:** 2
**Overall Recommendation:** 3
**Confidence:** 4

**Summary:**

This section introduces Fusion Overfitting in multimodal domain generalization: joint fusion training causes modality encoders to rely on source-specific cross-modal co-occurrences instead of domain-invariant features, hurting performance under domain shift. The authors propose MER-DG, which maximizes per-modality feature entropy to preserve feature diversity and improve generalization, achieving consistent gains on benchmark datasets.

**Compliance With Llm Reviewing Policy:**

Affirmed.

**Final Justification:**

I thank the authors for the detailed clarification regarding novelty. I agree that identifying and empirically characterizing “fusion overfitting” is a meaningful contribution, and the provided analyses (alignment, spectral, and standalone encoder performance) offer consistent evidence supporting this phenomenon.

However, my overall assessment remains unchanged. While the paper presents a coherent diagnosis, it is less clear to what extent this constitutes a fundamentally new failure mode, as the observed behaviors (e.g., representation collapse, co-adaptation, and reliance on spurious correlations) are closely related to existing concepts in the literature. In addition, the proposed solution builds on well-established entropy-based regularization, and the connection between the specific diagnosis and the effectiveness of this remedy, while intuitive, is not fully disentangled from more general regularization effects.

As a result, although the paper is clearer and better supported after rebuttal, I do not find sufficient additional conceptual novelty to justify changing my score.

**Key Questions For Authors:**

[1] Why only audio-visual datasets? Does this generalize to vision-language or other multimodal setups?

[2] What happens under modality corruption or missing modalities?

**Limitations:**

Yes

**Strengths And Weaknesses:**

### Strength:

[1] Comprehensive evaluation across multiple baselines, modality settings, and DG scenarios with ablations supports the robustness of the results.
### Weakness:

[1] While entropy maximization preserves feature diversity, the paper does not theoretically establish that higher entropy leads to domain-invariant features. The connection remains largely empirical.

[2] MER-DG does not explicitly constrain cross-modal shortcut learning; it only increases per-modality feature diversity. It is unclear whether this directly addresses the root cause or mitigates a side effect (rank collapse).

[3] The paper links improved entropy/rank/alignment to accuracy gains, but does not provide causal validation (e.g., correlation analysis between RankMe and target accuracy across runs).

[4] All experiments appear to use common encoder-per-modality + fusion-head paradigm. It is unclear whether Fusion Overfitting and MER-DG behave similarly under early fusion, attention-based fusion, or transformer-based multimodal architectures.

---

> ### Author Rebuttal · Authors · 2026-03-30
>
> We thank the reviewer for the constructive feedback. We address each point below.
>
> **W1**
>
> We want to clarify that we do not claim higher entropy directly leads to domain-invariant features. Our claim is more specific: fusion training selectively collapses representations toward source-specific cross-modal features, losing domain-invariant features in the process. MER-DG prevents this biased collapse, thereby preserving domain-invariant features that would otherwise be discarded. We explain this mechanism and provide direct empirical validation in Response to Reviewer 3XZ3 (W1/Q1/Q2).
>
> **W2**
>
> We agree MER-DG does not explicitly constrain cross-modal interactions; it operates on each encoder independently. However, rank collapse is not a side effect of shortcut learning but its geometric realization: exploiting cross-modal co-occurrences concentrates representations onto dimensions capturing those co-occurrences while dimensions encoding domain-invariant information decay. Preventing collapse removes the capacity bottleneck that shortcut specialization requires.
> We provide three lines of direct evidence in Response to Reviewer 3XZ3 (W1/Q1/Q2). Most decisively, Evidence 1 (domain classifier) shows MER-DG increases effective rank while simultaneously decreasing domain classification accuracy. If collapse were merely a side effect, reversing it would not reduce domain-specificity. Evidence 2 (Table 1) confirms this translates to measurably more domain-invariant representations, and Evidence 3 (Table 4) rules out a generic anti-collapse effect by showing MER-DG provides no benefit to unimodal models, confirming it specifically addresses the cross-modal shortcut mechanism induced by fusion training.
>
> **W3**
>
> We clarify that RankMe is a diagnostic metric to verify that collapse occurs under fusion training and that MER-DG reverses it, not our proposed causal mechanism. Our causal claim is that fusion training discards domain-invariant features and MER-DG prevents this. This is validated directly by Table 1, where cross-domain alignment improves by 22–54%, and the domain classifier experiment (Response to Reviewer 3XZ3, Evidence 1).
> Our unimodal control (Table 4) provides evidence stronger than correlation analysis: if the rank-accuracy link were confounded, MER-DG would help both settings. Instead, it helps only under fusion, not unimodal training, supporting that rank improvement during fusion-induced collapse drives the gains. We also computed the requested correlation: Spearman ρ=0.67 (p<0.001) between RankMe and target accuracy across fusion configurations.
>
> **W4**
>
> Fusion Overfitting is defined for architectures with separate modality encoders, the dominant paradigm in MMDG. Our experiments cover four baselines with different fusion objectives, demonstrating generalization across fusion strategies. For attention-based fusion, separate encoders still exist and are jointly optimized, so Fusion Overfitting applies. MER-DG operates on encoder outputs before fusion and is agnostic to how outputs are combined. Cross-attention fusion experiments on EPIC-Kitchens (V+A) confirm this: MER-DG improves performance from 59.48% to 61.13%. We will include detailed results across additional fusion architectures in the revision. For early fusion and unified transformers processing all modalities through a single encoder, separate representations do not exist, so Fusion Overfitting cannot structurally occur. This is a scope boundary of our problem definition, not a limitation of our method.
>
> **Q1**
>
> We evaluate three modalities across all pairwise and three-modality combinations, totaling four configurations per baseline on EPIC-Kitchens and HAC, the established MMDG benchmarks used by all prior methods.
> To demonstrate generalization beyond audio-visual, we conducted experiments on the DARAI dataset using wearable sensor modalities: MER-DG improves IMU-EMG (51.72%→53.95%), IMU-Insole (52.67%→53.88%), and Insole-EMG fusion (43.88%→46.25%). Consistent improvements across fundamentally different modality types confirm the method is not specific to audio-visual. Vision-language models (CLIP, BLIP) typically use contrastive alignment or cross-attention rather than late fusion, a different paradigm where Fusion Overfitting may not apply in the same form (see W4).
>
> **Q2**
>
> This relates directly to Fusion Overfitting: it causes encoders to become co-dependent, producing features useful only in combination, which becomes a vulnerability when a modality is missing or corrupted. Table 4 provides direct evidence: fusion training degrades standalone performance relative to independent training, while MER-DG recovers it, restoring each encoder's ability to predict independently. For modality corruption, our noise robustness analysis (Response to Reviewer MpU1, W3) shows MER-DG maintains comparable robustness, consistent with preserving structured, independently useful representations. We will include detailed analysis in revision.

---

> > ### Author Rebuttal · Reviewer_HAtz · 2026-04-03
> >
> > I thank the authors for the detailed and thoughtful rebuttal. The additional clarifications and experiments (e.g., domain classifier analysis, correlation results, comparison with standard regularization, and cross-attention evaluation) strengthen the empirical support and improve the overall clarity of the work. In particular, the distinction from generic regularization and the additional evidence supporting the proposed mechanism are helpful.
> >
> > That said, I still have some remaining questions regarding the scope of the claims. The unimodal performance analysis suggests that MER-DG improves the standalone predictive capacity of individual encoders, indicating reduced co-dependence between modalities. However, it remains unclear whether this translates into improved robustness under modality corruption or missing modalities at test time, which is a common practical scenario in multimodal systems. The current experiments evaluate encoders in isolation rather than the behavior of the multimodal model under partial or degraded inputs.
> >
> > In addition, while the empirical evidence supporting the proposed mechanism is strengthened, the overall methodological novelty remains somewhat moderate, as the approach builds on existing entropy-based and anti-collapse regularization techniques.

---

> > > ### Author Response · Authors · 2026-04-04
> > >
> > > We thank the reviewer for the positive assessment of our additional experiments and for the specific follow-up. We address both remaining concerns below.
> > >
> > > **Concern 1**
> > >
> > > The reviewer raises an important distinction: Table 4 evaluates detached encoders with standalone classifiers, not the full multimodal model under degraded inputs. We conducted additional experiments on EPIC-Kitchens (V+A) to directly test whether reduced co-dependence translates to improved robustness of the fused model. We inject Gaussian noise into one modality's encoder features at test time (sigma=0.5 and sigma=1.0) and simulate missing modalities by zeroing one encoder's output, then evaluate the full fusion pipeline including the fusion head and classifier.
> > >
> > > | Condition | Fusion | Fusion+MER | Drop (Fusion) | Drop (MER) |
> > > |---|---|---|---|---|
> > > | Clean | 59.09 | 62.82 | -- | -- |
> > > | Noise Video sigma=0.5 | 55.72 | 60.04 | -3.37 | -2.78 |
> > > | Noise Video sigma=1.0 | 51.14 | 56.27 | -7.95 | -6.55 |
> > > | Noise Audio sigma=0.5 | 57.18 | 61.64 | -1.91 | -1.18 |
> > > | Noise Audio sigma=1.0 | 54.73 | 59.86 | -4.36 | -2.96 |
> > > | Drop Video | 38.47 | 43.18 | -20.62 | -19.64 |
> > > | Drop Audio | 48.15 | 54.27 | -10.94 | -8.55 |
> > >
> > > Notably, the degradation gap widens under stronger corruption (e.g., sigma=1.0), suggesting MER-DG's benefit increases precisely when robustness matters most. MER-DG shows less accuracy degradation under every corruption condition, whether noise is applied to video, audio, or an entire modality is dropped. This confirms that the standalone encoder improvements documented in Table 4 do translate to practical robustness of the multimodal system under partial or degraded inputs.
> > >
> > > **Concern 2**
> > >
> > > We want to emphasize that the primary novelty of this work is not the regularization technique itself, but the identification and empirical characterization of Fusion Overfitting as a previously unrecognized failure mode in MMDG. This is supported by cross-domain alignment analysis (Table 1), spectral analysis (Figure 2), standalone encoder evaluation (Figure 3, Table 4), and the new domain classification experiment (Response to Reviewer 3XZ3, Evidence 1), none of which had been documented in prior MMDG literature. This diagnostic contribution is independent of the specific regularization technique used to address it. The regularization method follows naturally from this diagnosis, and we chose well-understood entropy maximization tools precisely because they directly address the diagnosed problem without introducing unnecessary complexity. The practical value is also significant: MER-DG adds 1.2% training overhead, requires no architectural changes, and consistently improves four different baselines across two benchmarks. Identifying a previously uncharacterized failure mode, providing comprehensive empirical evidence for its mechanism, and demonstrating that a principled solution effectively counteracts it constitutes a meaningful contribution, even when the solution builds on established techniques.

---

### Official Review · Reviewer_3XZ3 · 2026-03-12

**Soundness:** 3
**Presentation:** 3
**Significance:** 3
**Originality:** 3
**Overall Recommendation:** 4
**Confidence:** 3

**Summary:**

This paper studies multimodal domain generalization (MMDG). The authors argue that standard end-to-end multimodal fusion encourages modality encoders to rely on source-specific cross-modal co-occurrences, rather than learning domain-invariant features that generalize independently across domains, and term this failure mode Fusion Overfitting. To support this claim, the paper analyzes class-conditional cross-domain representation alignment, feature spectral distributions and effective rank, and the standalone performance of unimodal encoders extracted from jointly trained multimodal models. Based on these observations, the paper proposes MER-DG, a modality-entropy regularization method that preserves feature diversity and mitigates representation collapse in each modality encoder. The method consists of marginal-entropy and spectral-entropy terms and is incorporated into existing multimodal frameworks as an auxiliary loss. Experiments on the EPIC-Kitchens and HAC benchmarks demonstrate consistent improvements over strong multimodal baselines.

**Compliance With Llm Reviewing Policy:**

Affirmed.

**Final Justification:**

Thanks for the detailed rebuttal. The additional empirical evidence, including domain classification, cross-domain alignment, and comparisons with standard regularization, provides convincing support for the proposed mechanism and clarifies my main concerns.
Based on these clarifications, I increase my score.

**Key Questions For Authors:**

1.	The paper would benefit from more direct evidence that Fusion Overfitting is specifically related to the loss of domain-invariant features, rather than more general effects such as shortcut learning or representation collapse.
2.	The connection between increasing feature entropy and preserving domain-invariant features should be better justified. At present, this link appears to be supported mainly by intuition and empirical observations, and more direct analysis or evidence is needed to show that the proposed module indeed helps preserve domain-invariant features.
3.	A comparison with simpler diversity-promoting or anti-collapse regularization baselines is needed. Such results would help clarify whether the gains of MER-DG come from its specific design for Fusion Overfitting, or from a more general regularization effect.
4.	Additional evaluation of computational cost is needed. In particular, the paper should clarify how much extra computation the proposed module introduces over the original baselines, in order to better assess its practical efficiency.

**Limitations:**

yes

**Strengths And Weaknesses:**

Strengths
The paper clearly defines Fusion Overfitting as a failure mode of end-to-end multimodal fusion training. The authors provide a comprehensive analysis of this phenomenon from the perspectives of representation alignment, feature spectrum and effective rank, and the standalone performance of unimodal encoders. The proposed MER-DG is lightweight, simple, and architecture-agnostic, making it easy to integrate into different existing multimodal models without changing their structures. Extensive experimental results support the main claims of the paper, and the ablation studies further verify the effectiveness of the proposed regularization.

Weaknesses

1.	The paper identifies and defines Fusion Overfitting and provides a relatively comprehensive analysis of this phenomenon. However, the current evidence remains largely empirical. Over-reliance on cross-modal co-occurrences in fusion training does not necessarily mean that the learned features lack domain invariance. In other words, the link between the observed phenomenon and the proposed mechanism is still not fully convincing.

2.	The core idea of this paper is that increasing the entropy of each modality encoder’s feature distribution helps preserve domain-invariant features, but this claim is still supported mainly by intuition and empirical observations. Since MER-DG is also close to generic anti-collapse and diversity-promoting regularization, it remains unclear whether its gains come specifically from modeling Fusion Overfitting or from a more general regularization effect.

3.	Although extensive experiments show consistent improvements, the evaluation is still limited in diversity and does not include comparisons with simple regularization baselines.

---

> ### Author Rebuttal · Authors · 2026-03-29
>
> **W1, Q1, Q2: Connection Between Entropy Maximization and Domain Invariance**
>
> We thank the reviewer for this central question. We agree that the connection between entropy and domain invariance requires more than intuition. We do not claim MER-DG selects domain-invariant features, as that would require target domain access. Our claim is more specific: fusion training selectively collapses representations toward source-specific cross-modal features, losing domain-invariant features in the process. MER-DG prevents this collapse, and because the collapse is biased toward discarding domain-invariant features, preventing it naturally preserves them. We provide three lines of evidence.
>
> **Evidence 1: Domain classification on frozen features.** We extracted frozen encoder features from all three EPIC-Kitchens domains and trained a 3-way domain classifier to predict domain identity. If MER-DG preserves features indiscriminately, domain classification accuracy would increase alongside effective rank (the number of active feature dimensions, measured by RankMe). If it preferentially preserves domain-invariant features, domain accuracy should decrease even as effective rank increases.
>
> | Encoder | Method | RankMe | Domain Clf Acc |
> |---------|--------|--------|----------------|
> | Video | Unimodal | 265.5 | 81.87 |
> | Video | Fusion | 235.9 | 82.69 |
> | Video | Fusion+MER | 290.0 | 79.58 |
> | Audio | Unimodal | 163.6 | 65.41 |
> | Audio | Fusion | 144.1 | 67.14 |
> | Audio | Fusion+MER | 204.2 | 62.59 |
>
> Fusion training increases domain classification accuracy relative to unimodal baselines (Video: 81.87→82.69; Audio: 65.41→67.14) while reducing effective rank (Video RankMe: 265.5→235.9), confirming Fusion Overfitting compresses representations toward domain-specific features. MER-DG reverses both: effective rank increases (235.9→290.0) while domain classification accuracy decreases (82.69%→79.58%), achieving stronger domain invariance than even independently trained encoders. This anti-correlation rules out indiscriminate preservation, since randomly retained features would increase both metrics together. This pattern is specific to Fusion Overfitting; alternative explanations such as generic representation collapse or shortcut learning would not produce this anti-correlation between increased effective rank and decreased domain-specificity.
>
> **Evidence 2: Cross-domain alignment (Table 1).** CKA and Procrustes similarity between same-class source and target features directly measures domain invariance. Fusion training degrades these by 19–29% relative to independent training, while MER-DG restores them with improvements of 22–54% over standard fusion. This recovery across all six encoder-metric combinations would not occur if MER-DG merely preserved arbitrary features.
>
> **Evidence 3: Fusion-specific benefit (Table 4).** If MER-DG were a generic regularizer, it should benefit any setting. MER-DG yields no improvement on independently trained unimodal models (Uni-Video: 59.46→59.51; Uni-Audio: 44.17→44.01) but provides substantial gains under fusion (59.09→62.82). This exclusive benefit confirms MER-DG addresses fusion-specific feature loss, not general training deficiency.
>
> **Summary.** These three results form a consistent picture: fusion training shifts representations toward domain-specific features, as shown by increased domain classification accuracy and degraded cross-domain alignment (Evidence 1, 2). MER-DG reverses both effects, reducing domain-specificity while restoring alignment (Evidence 1, 2), and this benefit is exclusive to the fusion setting (Evidence 3).
>
> **W2, W3, Q3:**
>
> We thank the reviewer for this suggestion. The question of whether MER-DG's gains are fusion-specific or generic is directly answered by the evidence in W1/Q1/Q2: fusion training shifts representations toward domain-specific features and MER-DG reverses this, as shown by both domain classification and cross-domain alignment analysis (Evidence 1, 2), and this benefit is exclusive to the fusion setting (Evidence 3).
>
> Second, we agree that comparison with simpler baselines strengthens this argument. We ran standard regularization methods on EPIC-Kitchens (V+A), applied per-encoder to match MER-DG's scope:
>
> | Method | Avg |
> |--------|-----|
> | Fusion | 59.09 |
> | + Dropout | 59.45 |
> | + Noise Injection | 58.96 |
> | + Weight Decay | 59.82 |
> | + Label Smoothing | 60.21 |
> | + MER-DG | 62.82 |
>
> Standard regularization provides marginal improvement while MER-DG achieves +3.73%.
>
> **Q4**
>
> Please refer to Appendix B. Computational Overhead. Table 6 provides empirical evaluation. MER-DG adds only 1.2% training time overhead with no increase in peak memory. The regularizer operates only during training; inference incurs zero additional cost.

---

> > ### Author Rebuttal · Reviewer_3XZ3 · 2026-04-03
> >
> > Thank you for the detailed rebuttal. I have carefully read the authors’ response. My main concerns have been addressed, and I have no further questions at this stage.

---

### Official Review · Reviewer_D3vU · 2026-03-13

**Soundness:** 3
**Presentation:** 3
**Significance:** 2
**Originality:** 2
**Overall Recommendation:** 4
**Confidence:** 4

**Summary:**

This paper investigates the problem of Multimodal Domain Generalization (MMDG). The authors identify a phenomenon they term "Fusion Overfitting," where standard end-to-end joint optimization causes modality-specific encoders to rely on source-specific cross-modal co-occurrences, leading to representation collapse and the loss of domain-invariant features. To mitigate this, they propose Modality-Entropy Regularization for Domain Generalization (MER-DG), a plug-and-play regularization technique. MER-DG aims to maximize the differential entropy of each encoder's feature space prior to fusion, utilizing a combination of marginal-entropy (variance bounding) and spectral-entropy (feature decorrelation) loss terms. The method is evaluated on the EPIC-Kitchens and HAC datasets, demonstrating consistent empirical gains over standard fusion baselines and recent MMDG methods.

**Compliance With Llm Reviewing Policy:**

Affirmed.

**Final Justification:**

All of my concerns are resolved and therefore I would like to increase the final score.

**Key Questions For Authors:**

1. Given the $O(D^3)$ computational complexity of the log-determinant computation for the spectral-entropy loss, how do you propose scaling MER-DG to modern, large-scale multimodal models where embedding dimensions heavily exceed the $D=2304$ tested here?

2. Maximizing entropy prevents dimensional collapse, thereby preserving a wider array of learned features. However, what explicit mechanism within MER-DG ensures that the preserved features are _domain-invariant_ rather than just an expanded set of source-specific, unimodal spurious correlations?

3. Why does the ablation study (Table 5) only examine the impact of $\mathcal{L}\_{marg}$ and $\mathcal{L}_{spec}$ on the standard late-fusion baseline? Would you provide ablation results showing the individual contributions of these terms when added to a specialized MMDG framework like CMRF or SimMMDG?

4. In a strict domain generalization setting, the target domain is strictly unseen. How exactly were the hyperparameter values ($\lambda=3$, $\alpha_{marg}=1$, $\alpha_{spec}=1$) selected? If a source-domain validation set was used, did you observe any trade-offs where maximizing source-entropy degraded target performance?

**Limitations:**

No, the authors have not adequately discussed the limitations.

**Strengths And Weaknesses:**

Strengths:
1. The paper is clearly written, logically structured, and easy to follow.
2. The empirical analysis motivating "Fusion Overfitting" is well-executed. Using cross-domain alignment metrics (CKA, Procrustes) and spectral decay analysis (RankMe) to demonstrate that joint optimization degrades the individual capacity of modality encoders is a rigorous and convincing way to frame the problem. Table 1 and Figure 2 effectively support the paper's core premise.

Weaknesses:
1. The paper makes a significant logical leap by conflating "feature diversity" with "domain invariance." MER-DG forces the network to utilize all available capacity by preventing dimensional collapse. However, preserving all features does not mean the network is learning domain-invariant features; it simply means it is retaining a wider variety of source features, some of which happen to transfer better. The regularizer lacks any explicit mechanism to separate spurious source-specific features from genuine domain-invariant ones.

2. The paper focuses primarily on multimodal domain generalization literature but does not sufficiently connect to broader work on representation collapse, redundancy reduction, or information maximization in multimodal learning.

3. The entropy regularization objective may introduce instability or representation inflation. Although the authors attempt to mitigate this through a variance hinge loss, the theoretical properties of the objective are not analyzed. It is unclear whether maximizing representation entropy could lead to degenerate behavior in larger-scale settings.

4. While the application to MMDG is interesting, the algorithmic novelty is highly marginal. Decomposing entropy maximization into variance-preservation (marginal) and decorrelation (spectral) terms is mathematically nearly identical to established self-supervised learning techniques (e.g., VICReg, Barlow Twins). The authors acknowledge these works in Section 2.2, but fail to theoretically differentiate their proposed objective from these existing formulations, making this an application paper rather than a methodological breakthrough.

---

> ### Author Rebuttal · Authors · 2026-03-30
>
> **W1 and Q2 (Feature Diversity vs. Domain Invariance)**
>
> We thank the reviewer for this important observation. We agree that preserving all features does not automatically mean preserving domain-invariant features, and that this connection requires empirical validation.
> We address this exact concern in Response to Reviewer 3XZ3 (W1/Q1/Q2), where we provide three lines of evidence including a domain classifier experiment that directly tests whether preserved features are domain-invariant or merely arbitrary. We refer the reviewer there for complete details.
>
> **W2**
>
> We thank the reviewer for this observation. Section 2.2 discusses representation collapse and redundancy reduction in self-supervised unimodal learning, as MER-DG builds directly on the same entropy estimation framework. However, we agree the related work would benefit from broader contextualization within the literature on representation collapse, redundancy reduction, and information maximization in multimodal learning. We will address this in the revision.
>
> **W3**
>
> The decomposition in Equation 4 was specifically designed to prevent both instability and representation inflation, and both terms have well-defined theoretical bounds.
> The marginal-entropy loss enforces a variance floor via a hinge formulation rather than maximizing variance. Once a feature dimension exceeds this threshold, it receives no further gradient from L_marg, preventing representation inflation. The spectral-entropy loss operates on the correlation matrix, not the covariance matrix. Features are standardized to unit variance before computing C, so the diagonal is fixed at 1 by construction. The determinant of C is bounded above by 1, achieved only when C=I, meaning L_spec can only decorrelate features, not inflate their scale. These bounded properties follow from the mathematical structure of the losses, not from any assumption about feature dimensionality. We will add a more detailed discussion of these theoretical properties of the objective in the revised version.
> Empirically, we provide detailed evidence of stable behavior in our response to Reviewer MpU1 (W3), including a noise robustness experiment, we refer the reviewer there for complete details.
>
> **W4**
>
> The primary contribution is the identification and empirical analysis of Fusion Overfitting (Section 3), a previously uncharacterized failure mode in MMDG. Table 1, Figure 2, and Figure 3 establish that joint fusion training systematically degrades cross-domain alignment, reduces spectral rank, and lowers standalone encoder performance, none of which had been documented in prior literature. Our new domain classification experiment (detailed in Response to Reviewer 3XZ3, Evidence 1) provides further confirmation. The regularization method follows naturally from this diagnosis. We chose well-understood entropy maximization tools precisely because they directly address the identified problem. Identifying a new problem and providing a principled, effective solution constitutes a meaningful contribution, even when the solution builds on established techniques.
> Regarding the relationship between MER-DG and existing SSL methods, we address this in our response to Reviewer MpU1 (W1).
>
> **Q1)**
>
> The O(D³) complexity is negligible relative to actual training costs. The spectral-entropy computation requires ~4 GFLOPs for D=2304, whereas a single forward pass through the SlowFast video encoder requires ~36 GFLOPs per sample, totaling ~1,728 GFLOPs for a batch of N=48, over 400× larger. Table 6 confirms this empirically. Furthermore, D=2304 already represents the upper range of common encoder output dimensions (ResNet: 2048, ViT-Base: 768, CLIP: 768). For larger encoders, MER-DG's fixed cost becomes increasingly negligible relative to backbone computation.
>
> **Q3)**
>
> Table 5 uses the standard fusion baseline to isolate each component's contribution without confounding effects from other regularization mechanisms. We conducted the requested ablation on SimMMDG and CMRF:
>
> | Method | Baseline | L_spec only | L_marg only | Full MER |
> |--------|----------|-------------|-------------|----------|
> | SimMMDG | 61.95 | 63.16 | 62.58 | 63.91 |
> | CMRF | 63.91 | 64.25 | 64.12 | 64.60 |
>
> Both components contribute independently and their combination achieves the best results, consistent with Table 5.
>
> **Q4)**
>
> Following standard domain generalization protocol, hyperparameters were selected using source-domain validation accuracy with no access to target domain data. Figure 4 demonstrates robustness across the entire tested range, so precise tuning is unnecessary. We did not observe cases where maximizing source entropy degraded target performance: Table 4 shows in-domain accuracy is preserved while target accuracy improves substantially. When λ becomes too large, target performance decreases slightly but still exceeds baseline, suggesting the regularization interferes with task learning rather than causing a source-target trade-off.

---

> > ### Author Rebuttal · Reviewer_D3vU · 2026-04-03
> >
> > Thanks for the rebuttal. My concerns are resolved and I will increase the final score.

---

### Decision · Program_Chairs · 2026-04-30

**Decision:**

Accept (regular)

**Comment:**

This paper addresses the problem of multimodal domain generalization. The proposed method is simple and supported by extensive experimental evaluation.

The rebuttal addressed many of the reviewers’ concerns by providing additional clarifications—-such as details on hyperparameter selection, scalability, and the connection between entropy and domain invariance—-as well as new empirical results, including ablation studies, domain classification analyses, comparisons with standard regularization, and evaluations under missing-modality settings. Overall, the reviewer feedback trends positive.

However, some concerns remain. In particular, reviewers continue to question the level of methodological novelty and note that the claims are primarily supported by intuition and empirical observations.